# Beyond Modality Collapse: Representations Blending for Multimodal Dataset Distillation

**Xin Zhang**[1,2]    **Ziruo Zhang**[3]    **Jiawei Du**[1,2]    **Zuozhu Liu**[4]    **Joey Tianyi Zhou**[1,2 ✉]

[1]Centre for Frontier AI Research, Agency for Science, Technology and Research, Singapore
[2]Institute of High Performance Computing, Agency for Science, Technology and Research, Singapore
[3]National University of Singapore, Singapore    [4]Zhejiang University, China
{zhangx7, dujw, Joey_Zhou}@a-star.edu.sg
ziruo.z@u.nus.edu    zuozhuliu@intl.zju.edu.cn

## Abstract

Multimodal Dataset Distillation (MDD) seeks to condense large-scale image-text datasets into compact surrogates while retaining their effectiveness for cross-modal learning. Despite recent progress, existing MDD approaches often suffer from *Modality Collapse*, characterized by over-concentrated intra-modal representations and enlarged distributional gap across modalities. In this paper, for the first time, we identify this issue as stemming from a fundamental conflict between the over-compression behavior inherent in dataset distillation and the cross-modal supervision imposed by contrastive objectives. To alleviate modality collapse, we introduce **RepBlend**, a novel MDD framework that weakens overdominant cross-modal supervision via representation blending, thereby significantly enhancing intra-modal diversity. Additionally, we observe that current MDD methods impose asymmetric supervision across modalities, resulting in biased optimization. To address this, we propose symmetric projection trajectory matching, which synchronizes the optimization dynamics using modality-specific projection heads, thereby promoting balanced supervision and enhancing cross-modal alignment. Experiments on Flickr-30K and MS-COCO show that RepBlend consistently outperforms prior state-of-the-art MDD methods, achieving significant gains in retrieval performance (e.g., +9.4 IR@10, +6.3 TR@10 under the 100-pair setting) and offering up to $6.7\times$ distillation speedup. Our code is publicly available at https://github.com/zhangxin-xd/RepBlend.

## 1 Introduction

The unprecedented expansion of large-scale datasets has catalyzed recent breakthroughs in deep learning [6, 2, 1], but has also introduced considerable storage and computational overhead [20, 22]. Thus, reducing dataset size to streamline the development process has emerged as an important research focus. Among various solutions, Dataset Distillation (DD) [50] has emerged as a compelling strategy, achieving high compression ratios by synthesizing a compact surrogate dataset that approximates the training efficacy of the original dataset. The effectiveness of DD has been demonstrated across various modalities, including images [4, 57], text [30, 32], videos [11, 51], and graphs [29, 58]. These unimodal successes motivate its extension to increasingly prominent multimodal scenarios [37, 28, 35, 5].

The pioneering effort in multimodal dataset distillation (MDD) is MTT-VL [53], which first validates the feasibility of extending existing vanilla DD techniques to the image-text setting. Building on this baseline, LoRS [55] further proposes to mine cross-modal similarity to calibrate the supervision

---

✉ Corresponding author.

39th Conference on Neural Information Processing Systems (NeurIPS 2025).

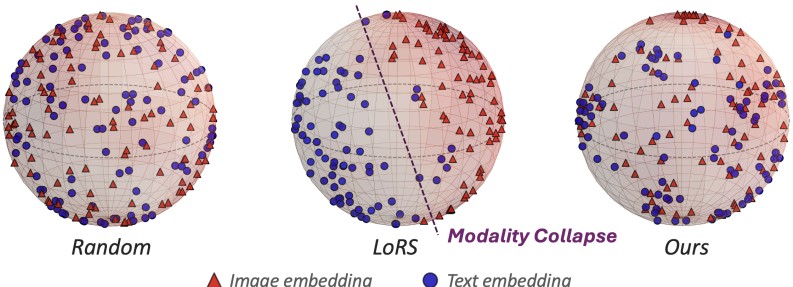

*Random*      *LoRS*    **Modality Collapse**     *Ours*

▲ *Image embedding*     ● *Text embedding*

Figure 1: Multimodal embedding distributions across various distillation methods. We extract image and text embeddings from a finetuned CLIP [37] and project them into a shared representation space using DOSNES [31]. Red triangles and blue circles denote image and text embeddings, respectively. **Left**: Embeddings from randomly sampled data in the original dataset exhibit a well-spread and modality-aligned distribution. **Middle**: The distilled dataset generated by a SOTA MDD method (LoRS [55]) suffers from *Modality Collapse*, where image and text embeddings are poorly aligned and concentrated in separate regions. **Right**: Our method effectively mitigates modality collapse, resulting in a distribution with improved cross-modal alignment and higher representational diversity.

from matched and mismatched pairs, thereby achieving better adaptation to high-variance image-text data. Despite achieving promising results, existing studies remain confined to the data structure level, without probing the underlying conflict between DD and contrastive learning. Specifically, to prevent significant performance deterioration, vanilla DD prioritizes capturing representative features under limited distillation budgets, often sacrificing diversity and distributional coverage [14, 18, 15]. While this compromise is tolerable in unimodal classification tasks, naively applying such strategies to multimodal contrastive learning, which places great importance on instance-level discriminability, inevitably leads to *Modality Collapse*. As illustrated in Figure 1 (middle), the distilled dataset exhibits pronounced intra-modality aggregation and inter-modality separation.

This modality collapse leads to two critical issues. First, *it induces excessive intra-modal similarity*, where embeddings within each modality become increasingly concentrated as distillation progresses. This over-concentration gradually suppresses representational diversity, making semantically distinct instances harder to separate, and eroding the fine-grained discrimination ability within each modality. Second, *it widens the inter-modal gap*, resulting in a large divergence between the feature distributions of different modalities. Insufficient cross-modal interaction fragments the embedding spaces and weakens semantic alignment, compromising the correct matching of positive pairs and the separation of negative pairs across modalities.

Recognizing these limitations, we propose **RepBlend**, a novel framework for MDD aimed at alleviating modality collapse. First, we theoretically identify that the collapse results from the joint effect of the over-compressive nature of DD, where optimization converges toward a small set of dominant features, and the cross-modal contrastive supervision, which further reinforces this convergence, leading to intra-modal collapse. To address this issue, RepBlend introduces Representation Blending within each modality to weaken the overly strong cross-modal supervision, thereby promoting intra-modal diversity. Furthermore, we observe that existing MDD approaches exhibit asymmetric supervision between modalities, with the image branch receiving significantly weaker update signals than the text branch. To address this, we propose Symmetric Projection Trajectory Matching, a mechanism that aligns the optimization trajectories of both projection heads, thereby enhancing cross-modal alignment and improving overall distillation efficiency. Extensive evaluations on Flickr-30K and MS-COCO demonstrate that RepBlend consistently surpasses existing MDD methods. Notably, under the 100-pair setting on Flickr-30K, it achieves improvements of +9.4 in IR@10 and +6.3 in TR@10, along with a $6.7\times$ distillation speedup over the SOTA baseline. Beyond these benchmarks, RepBlend also exhibits strong generalization to other multimodal scenarios, such as audio-text.

Our contributions are summarized as follows:

- For the first time, we identify the modality collapse issue in current MDD solutions, where the distilled dataset exhibits high intra-modal similarity and a large inter-modal gap. Through theoretical analysis, we attribute this to a mutually reinforcing effect between the over-compression behavior of dataset distillation and the cross-modal supervision enforced by contrastive objectives.

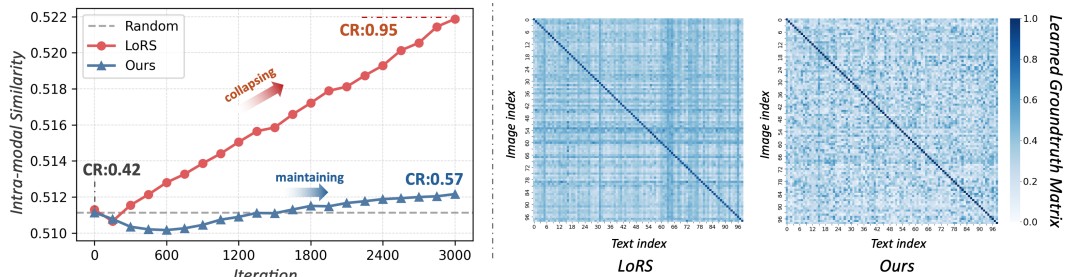

Figure 2: **Left**: Increasing intra-modal similarity as distillation progresses. We run optimization for 3000 iterations and track the intra-modal cosine similarity, which increases from 0.512 to 0.522 (red curve). Though small in magnitude, this rise leads to a more than twofold increase in concentration ratio (CR)[2] due to the high dimensionality of the embedding space. **Right**: Modality collapse undermines the effectiveness of learned soft cross-modal correspondence. The non-matching image-text pairs exhibit nearly uniform similarity scores, forming horizontal and vertical stripes.

- We propose Representation Blending to mitigate modality collapse by weakening the overly strong cross-modal supervision and enhancing intra-modal representational diversity. Furthermore, we introduce Symmetric Projection Trajectory Matching to enable more balanced multimodal distillation, which not only strengthens cross-modal alignment but also improves overall distillation efficiency.

## 2   Preliminaries and Related Works

Dataset Distillation (DD) [50] aims to synthesize a compact surrogate dataset that emulates the key properties of the original large-scale dataset. These properties include distributional characteristics, such as feature-level statistics [60, 48, 49] and batch normalization parameters [57, 42, 15], and training dynamics, including gradients [61, 59] and optimization trajectories [4, 7, 14, 18, 25]. While DD achieves promising results on unimodal benchmarks, extending it to multimodal scenarios remains challenging due to unique data structures and learning strategies [53, 55]. We begin by formalizing the problem of Multimodal Dataset Distillation (MDD).

**Problem Formulation.**   Given a large-scale image-text dataset $\mathcal{D} = \{(\boldsymbol{x}_i, \boldsymbol{\tau}_i), \boldsymbol{y}_i\}_{i=1}^{|\mathcal{D}|}$, where $\boldsymbol{x}_i \in \mathbb{R}^{d_{\text{img}}}$ and $\boldsymbol{\tau}_i \in \mathbb{R}^{d_{\text{text}}}$ denote the $i$-th image and its paired caption representation[1], and each pair is independently sampled from a natural data distribution $\mathcal{P}$. Each $\boldsymbol{y}_i \in \{0, 1\}^{|\mathcal{D}|}$ is a one-hot vector indicating the correspondence between $\boldsymbol{x}_i$ and the caption set $\{\boldsymbol{\tau}_j\}_{j=1}^{|\mathcal{D}|}$, with the $i$-th entry activated. Similar to DD, MDD also aims to minimize the loss on original dataset using the model trained on its distilled synthetic counterpart $\mathcal{S} = \{(\tilde{\boldsymbol{x}}_i, \tilde{\boldsymbol{\tau}}_i), \tilde{\boldsymbol{y}}_i\}_{i=1}^{|\mathcal{S}|}$:

$$\mathcal{S}^* = \arg\min_{\mathcal{S}} \mathbb{E}_{(\boldsymbol{x}, \boldsymbol{\tau}) \sim \mathcal{P}} [\mathcal{L}(f_{\boldsymbol{\theta}_{\mathcal{S}}}(\boldsymbol{x}, \boldsymbol{\tau}), \boldsymbol{y})] \quad \text{s.t.} \quad \boldsymbol{\theta}_{\mathcal{S}} = \arg\min_{\boldsymbol{\theta}} \mathbb{E}_{(\tilde{\boldsymbol{x}}, \tilde{\boldsymbol{\tau}}) \sim \mathcal{S}} [\mathcal{L}(f_{\boldsymbol{\theta}}(\tilde{\boldsymbol{x}}, \tilde{\boldsymbol{\tau}}), \tilde{\boldsymbol{y}})], \quad (1)$$

where $|\mathcal{S}| \ll |\mathcal{D}|$, and $\mathcal{L}$ denotes the contrastive learning loss. The model $f_{\boldsymbol{\theta}}(\cdot)$ represents a CLIP-style network parameterized by $\boldsymbol{\theta}$. Each distilled sample consists of a synthetic image-text pair $(\tilde{\boldsymbol{x}}_i, \tilde{\boldsymbol{\tau}}_i)$, where $\tilde{\boldsymbol{x}}_i \in \mathbb{R}^{d_{\text{img}}}$ and $\tilde{\boldsymbol{\tau}}_i \in \mathbb{R}^{d_{\text{text}}}$, accompanied by a learned soft label $\tilde{\boldsymbol{y}}_i$.

**MDD vs. Vanilla DD.**   According to the Equation 1, the generalization from vanilla DD to MDD involves two key modifications: 1) introducing soft ground-truth vectors $\tilde{\boldsymbol{y}}_i$, and 2) optimizing under a contrastive learning loss $\mathcal{L}$ for image-text alignment. While learning soft labels is common in vanilla DD [7], optimizing $\tilde{\boldsymbol{y}}_i$ in MDD is more challenging, as both image and text representations are updated simultaneously. Besides, in practice, the contrastive loss $\mathcal{L}$ is typically instantiated as InfoNCE [33], extended InfoNCE (eNCE), or weighted BCE (wBCE) [55], all aiming to strengthen positive alignments while penalizing mismatched pairs. However, these extensions only make the multimodal adaptation feasible, overlooking the essence of dataset distillation: effective information

---

[1]Given the discrete nature of text, all subsequent analysis is conducted in the representation space, while images remain processed in the pixel space. Here, $d_{\text{img}} = W \times H \times 3$ and $d_{\text{text}} = 768$ (for BERT [10]).

[2]CR measures how tightly the features are clustered, based on how much of the hypersphere is covered at the given cosine similarity. (Refer to Appendix C for more calculation details).

condensation. More specifically, they prioritize cross-modal alignment, while failing to preserve intra-modal diversity and discriminability under severe data compression.

# 3 Methodology

In this section, we introduce **RepBlend**, a novel approach for MDD. We begin by identifying the phenomenon of *Modality Collapse*, which emerges when vanilla DD methods are naively applied to multimodal settings. Through theoretical and empirical analysis, we uncover its underlying causes. To address this issue, we propose Representation Blending to enhance intra-modal diversity. In addition, we introduce Symmetric Projection Trajectory Matching, which balances the distillation process across modalities and further strengthens cross-modal alignment. The overall pipeline of RepBlend is outlined in Algorithm 1.

## 3.1 Modality Collapse

LoRS [55] is a representative MDD method built upon Equation 1, where $\mathcal{L}$ is defined as:

$$\mathcal{L}_{\text{wBCE}}^{\mathcal{B}} = \sum_{i,j}^{|\mathcal{B}|} w_{ij} \cdot \ell\left(\tilde{\boldsymbol{y}}_{ij}, \sigma\left(\hat{\boldsymbol{y}}_{ij}/\gamma\right)\right), \quad w_{ij} = \frac{\mathbb{I}[\tilde{\boldsymbol{y}}_{ij} > \beta]}{|\{(i,j) : \tilde{\boldsymbol{y}}_{ij} > \beta\}|} + \frac{\mathbb{I}[\tilde{\boldsymbol{y}}_{ij} \le \beta]}{|\{(i,j) : \tilde{\boldsymbol{y}}_{ij} \le \beta\}|}. \quad (2)$$

Here, $\mathcal{B} \subset \mathcal{S}$ denotes a sampled batch. $\hat{\boldsymbol{y}}_{ij}$ represents the cosine similarity between the normalized image and text embeddings, where $\tilde{\boldsymbol{x}}_i' = \text{Normalize}(f^{\text{imgE}}(\tilde{\boldsymbol{x}}_i))$[3] and $\tilde{\boldsymbol{\tau}}_j' = \text{Normalize}(f^{\text{textP}}(\tilde{\boldsymbol{\tau}}_j))$, with $f^{\text{imgE}}(\cdot)$ and $f^{\text{textP}}(\cdot)$ denoting the image encoder and text projection head, respectively. The threshold $\beta$ is used to determine positive and negative pairs, $\sigma(\cdot)$ denotes the sigmoid function, and $\gamma$ is the temperature. $\ell(\cdot, \cdot)$ refers to the binary cross-entropy loss. While this supervision primarily aims to mine cross-modal relationships, it inadvertently reinforces intra-modal similarities, ultimately leading to *Modality Collapse*, as shown in Figure 1, where instances within each modality excessively concentrate. Without loss of generality, the following analysis focuses on the image modality.

***Proposition: Cross-modal supervision reinforces intra-modal similarity.*** During dataset distillation, if $\{\tilde{\boldsymbol{x}}_n, \tilde{\boldsymbol{\tau}}_n\}$ and $\{\tilde{\boldsymbol{x}}_m, \tilde{\boldsymbol{\tau}}_m\}$ exhibit some non-negligible similarity, i.e., $\tilde{\boldsymbol{y}}_{nm} \approx \tilde{\boldsymbol{y}}_{mn} > \beta$, then the direction of their subsequent updates $\frac{\partial \mathcal{L}}{\partial \tilde{\boldsymbol{x}}_n'} \frac{\partial \mathcal{L}}{\partial \tilde{\boldsymbol{x}}_m'}$ is determined by

$$\frac{w_{nm}w_{mn}}{\gamma^2}[\sigma(\hat{\boldsymbol{y}}_{nm})/t - \tilde{\boldsymbol{y}}_{nm}][\sigma(\hat{\boldsymbol{y}}_{mn})/t - \tilde{\boldsymbol{y}}_{mn}]\tilde{\boldsymbol{\tau}}_m'^{\top}\tilde{\boldsymbol{\tau}}_n', \quad (3)$$

which indicates that the optimization is guided by positive pairs $\tilde{\boldsymbol{\tau}}_m'^{\top}\tilde{\boldsymbol{\tau}}_n'$, promoting concentration in similar directions. A detailed derivation is provided in Appendix B. When distilling a large dataset into a compact one, the optimization process tends to be dominated by a few salient features [9, 15, 18, 43]. Once this convergence trend emerges, cross-modal supervision further reinforces it: modality-specific diversity is implicitly suppressed, and intra-modal representations are increasingly aligned toward a limited set of dominant directions. As illustrated in Figure 2 (left), *the intra-modal similarity consistently increases throughout the distillation process.*

In addition to the aggravated intra-modal similarity, modality collapse also exacerbates the cross-modal representation gap, as features from each modality become increasingly centralized within

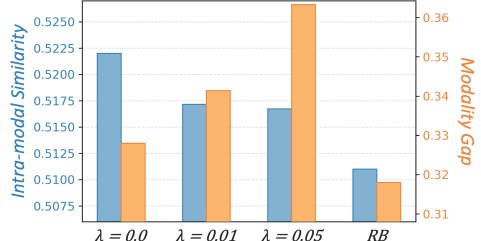

Figure 3: As the noise level $\lambda$ increases, intra-modal similarity (blue bars) shows a slight decline, while the modality gap (yellow bars) rises markedly. In contrast, our representation blending (RB) leverages in-distribution samples to simultaneously reduce intra-modal similarity and inter-modal gap, effectively mitigating modality collapse during distillation.

compact regions of the shared embedding space. Consequently, the similarities between non-matching image-text pairs converge toward a uniform distribution. Such behavior undermines the utility of soft label distributions, which are designed to encode fine-grained relational information beyond

---

[3]In LoRS [55], no image projection head is used.

the binary supervision provided by one-hot labels. As illustrated in Figure 2 (right), *non-diagonal similarity values exhibit a near-uniform pattern*, where image embeddings produce nearly constant similarity scores across all non-matching text embeddings (manifesting as horizontal stripes), and vice versa for text samples (vertical stripes).

## 3.2 Mitigating Modality Collapse via Representation Blending

As analyzed in Equation 3, modality collapse arises from overly strong cross-modal supervision, which implicitly encourages intra-modal concentration and undermines representational diversity. To alleviate this constraint, one potential approach is to inject directional signals that deviate from $\tilde{\tau}'_m$ and $\tilde{\tau}'_n$. To empirically validate this hypothesis and explore a viable remedy, we conduct a controlled perturbation experiment on Flickr-30K [36]. In particular, we adopt two key metrics following [26]: the intra-modal similarity (Sim) and the modality gap (Gap), defined as,

$$\texttt{Sim} = \frac{1}{|\mathcal{S}|(|\mathcal{S}|-1)} \sum_{i \neq j}^{|\mathcal{S}|} \tilde{\boldsymbol{x}}'^{\top}_i \tilde{\boldsymbol{x}}'_j, \quad \texttt{Gap} = \frac{1}{|\mathcal{S}|} \| \sum_{i=1}^{|\mathcal{S}|} \tilde{\boldsymbol{x}}'_i - \sum_{j=1}^{|\mathcal{S}|} \tilde{\boldsymbol{\tau}}'_j \|_2. \tag{4}$$

We inject Gaussian noise into the text representations,

$$\tilde{\boldsymbol{\tau}}'^{+\text{noise}}_m = \text{Normalize}\left(f^{\text{textP}}((1-\lambda)\tilde{\boldsymbol{\tau}}_m + \lambda\vec{\Delta}_m)\right), \quad \tilde{\boldsymbol{\tau}}'^{+\text{noise}}_n = \text{Normalize}\left(f^{\text{textP}}((1-\lambda)\tilde{\boldsymbol{\tau}}_n + \lambda\vec{\Delta}_n)\right),$$

where $\vec{\Delta}_m$ and $\vec{\Delta}_n$ are independently sampled random noise from $\mathcal{N}(0,1)$, and $\lambda$ controls the noise level. We evaluate Sim and Gap under varying levels of $\lambda$. As shown in Figure 3, a slight increase in noise reduces intra-modal similarity (blue bars), indicating enhanced modality-specific diversity. These results support our hypothesis that perturbing in the representation space can effectively counteract modality concentration.

However, as noise level continues to grow, the injected perturbation begins to introduce semantically meaningless signals, which hinders cross-modal alignment. This is evidenced by the growing modality gap (yellow bars), accompanied by a performance drop of 1.9% in IR@1 and 2.1% in TR@1 at $\lambda = 0.01$ under 100 distilled pairs on Flickr-30K dataset. To mitigate this issue, we propose replacing the random perturbation with a structure-preserving variant using in-distribution samples. Specifically, we blend representations from different synthetic instances:

$$\tilde{\boldsymbol{\tau}}'^{\text{blend}}_m = \text{Normalize}\left(f^{\text{textP}}((1-\lambda)\tilde{\boldsymbol{\tau}}_m + \lambda\tilde{\boldsymbol{\tau}}_i)\right), \quad \tilde{\boldsymbol{\tau}}'^{\text{blend}}_n = \text{Normalize}\left(f^{\text{textP}}((1-\lambda)\tilde{\boldsymbol{\tau}}_n + \lambda\tilde{\boldsymbol{\tau}}_j)\right), \tag{5}$$

where $1 \leq i, j \leq |\mathcal{S}|$. This operation resembles the idea of $\text{MixUp}$, but is applied in the representation space. As shown in the last group of Figure 3, we can maintain a low level of intra-modal similarity and small modality gap. Note that although here we illustrate the formulation on text, the same operation is also applied to image side in practice.

## 3.3 Enhancing Cross-modal Alignment via Symmetric Projection Trajectory Matching

In prior MDD practices, methods such as MTT-VL [53] and LoRS [55] follow a de facto protocol wherein the text encoder is frozen and the image projection layer is omitted. The image encoder and the text projection head are trained to generate expert trajectories for distillation. In this setup, the image encoder is initialized with pretrained weights from ImageNet-1K [8], while the text projection head is trained from scratch. This design is motivated by two key considerations: 1) the prohibitive computational and memory cost of optimizing and storing expert trajectories for large-scale text encoders such as BERT [10]; and 2) the fact that text distillation operates in the representation space, where supervision is applied only through the projection head, thus, matching at the encoder level cannot propagate supervision to the representation space. LoRS [55] minimize the objective in Equation 1 through trajectory matching, which is formulated as follows:

$$\tilde{\boldsymbol{x}}^*, \tilde{\boldsymbol{\tau}}^*, \tilde{\boldsymbol{y}}^* = \underset{\tilde{\boldsymbol{x}}, \tilde{\boldsymbol{\tau}}, \tilde{\boldsymbol{y}}}{\arg\min} \left( \left\| \boldsymbol{\theta}^{t+T}_{\mathcal{S}_{\text{imgE}}} - \boldsymbol{\theta}^{t+M}_{\mathcal{D}_{\text{imgE}}} \right\|_2^2 + \left\| \boldsymbol{\theta}^{t+T}_{\mathcal{S}_{\text{textP}}} - \boldsymbol{\theta}^{t+M}_{\mathcal{D}_{\text{textP}}} \right\|_2^2 \right) / \left( \left\| \boldsymbol{\theta}^{t}_{\mathcal{D}_{\text{imgE}}} - \boldsymbol{\theta}^{t+M}_{\mathcal{D}_{\text{imgE}}} \right\|_2^2 + \left\| \boldsymbol{\theta}^{t}_{\mathcal{D}_{\text{textP}}} - \boldsymbol{\theta}^{t+M}_{\mathcal{D}_{\text{textP}}} \right\|_2^2 \right),$$

where $\boldsymbol{\theta}^{t+T}_{\mathcal{S}\text{imgE}}$ and $\boldsymbol{\theta}^{t+T}_{\mathcal{S}\text{textP}}$ denote the $T$-step finetuned weights of the image encoder and text projection head using $\mathcal{S}$, initialized from $\boldsymbol{\theta}^{t}_{\mathcal{D}_{\text{imgE}}}$ and $\boldsymbol{\theta}^{t}_{\mathcal{D}_{\text{textP}}}$, respectively. The objective is to align the $T$-step synthetic trajectory with the $M$-step real trajectory by minimizing the $\ell_2$ distance between their terminal weights, given the same initialization.

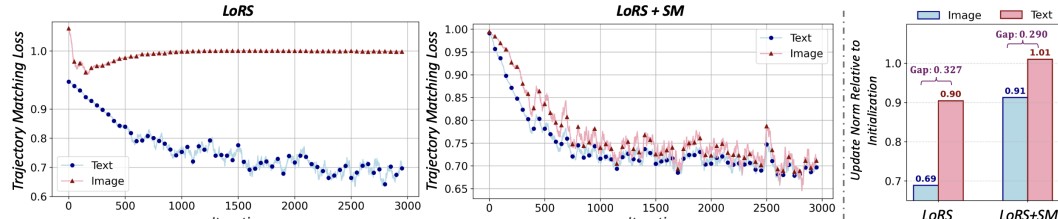

Figure 4: Current MDD methods adopt asymmetric distillation. **Left**: The loss on the image side shows much smaller variation than that of the text side, fluctuating mildly around 1.0 without notable reduction. **Right**: The update norm relative to initialization is significantly lower for the image modality in LoRS (0.69) compared to the text modality (0.90), suggesting insufficient representation transfer. The update norm is computed in the shared representation space for both modalities. After incorporating symmetric matching (SM), both image and text modalities exhibit more balanced and synchronized update dynamics, leading to more effective cross-modal alignment (reduced Gap).

However, the aforementioned trajectory matching is asymmetric. As shown in Figure 4 (left), the trajectory matching losses of the image and text modalities exhibit divergent trends: the text-side loss decreases steadily, whereas the image-side loss quickly plateaus and remains relatively high. This is primarily because the image encoder contains significantly more parameters than the text projection head, thus, even small per-parameter errors can accumulate into a large overall mismatch. This imbalance is further evidenced in Figure 4 (right), the norm of updates relative to initialization for the image modality is significantly smaller than that of the text, indicating insufficient distillation on the image side. While the representation blending introduced in Section 3.2 helps narrow the modality gap, its effect is still constrained by the inherently asymmetric distillation. To address this imbalance and further enhance cross-modal alignment, we propose a symmetric distillation strategy by matching trajectories of projection head for both modalities:

$$\tilde{\boldsymbol{x}}^*, \tilde{\boldsymbol{\tau}}^*, \tilde{\boldsymbol{y}}^* = \arg\min_{\tilde{\boldsymbol{x}}, \tilde{\boldsymbol{\tau}}, \tilde{\boldsymbol{y}}} \left( \left\| \boldsymbol{\theta}_{\mathcal{S}_{\text{imgP}}}^{t+T} - \boldsymbol{\theta}_{\mathcal{D}_{\text{imgP}}}^{t+M} \right\|_2^2 + \left\| \boldsymbol{\theta}_{\mathcal{S}_{\text{textP}}}^{t+T} - \boldsymbol{\theta}_{\mathcal{D}_{\text{textP}}}^{t+M} \right\|_2^2 \right) / \left( \left\| \boldsymbol{\theta}_{\mathcal{D}_{\text{imgP}}}^{t} - \boldsymbol{\theta}_{\mathcal{D}_{\text{imgP}}}^{t+M} \right\|_2^2 + \left\| \boldsymbol{\theta}_{\mathcal{D}_{\text{textP}}}^{t} - \boldsymbol{\theta}_{\mathcal{D}_{\text{textP}}}^{t+M} \right\|_2^2 \right). \quad (6)$$

Here, the image encoder is initialized with ImageNet-1K pretrained weights and kept frozen. While the added image projection head incurs slight computational overhead, it enables projection-based matching that significantly enhances the overall efficiency of the distillation process (as discussed in Section 4.4). As shown in Figure 4, symmetric projection matching leads to a more consistent decrease in loss for both image and text branches. Moreover, the increased magnitude of updates suggests stronger supervision signals across modalities, resulting in a more balanced and effective distillation process. With symmetric distillation, the modality gap is further narrowed from 0.318 (in Figure 3) to 0.290, indicating enhanced cross-modal alignment.

## 4 Experiments

In this section, we conduct extensive experiments on multiple benchmark datasets to demonstrate the effectiveness of the proposed RepBlend framework. We first present the experimental setup, including the datasets, baseline methods, and implementation details. The main results are summarized in Table 1, Table 2, and Table 3. In addition, we also provide detailed ablation studies to evaluate the individual contribution of each component. All experiments are conducted using two NVIDIA RTX 3090 GPUs and one NVIDIA H100 GPU.

### 4.1 Experimental Setup

**Datasets and Networks.** We evaluate our method on two widely-used image captioning datasets: Flickr-30K [36] and MS-COCO [27], which contain approximately 31k and 123k images respectively, with each image paired with five human-annotated captions. For the image encoder, we experiment with NFNet [3], RegNet [38], ResNet-50 [19], and ViT [12]. For the text encoder, we consider both BERT [10] and DistilBERT [40]. To further demonstrate the generalizability of our approach across modalities, we extend our evaluation to the AudioCaps [23] audio-text benchmark, utilizing EfficientAT [41] as the audio encoder. Model performance is primarily evaluated using Recall at K (R@K) in cross-modal retrieval tasks. Given a query from one modality, we retrieve the top-$K$ most similar samples from the other modality and measure the retrieval accuracy. We denote text-to-image retrieval as IR@K, and image-to-text retrieval as TR@K.

---

**Algorithm 1** Blending Representations to Mitigate Modality Collapse in MDD

---

**Require:** Original large dataset $\mathcal{D}$; CLIP-style network $\{f^{\text{imgE}}, f^{\text{textE}}, f^{\text{imgP}}, f^{\text{textP}}\}$; real trajectories set $\Theta_{\mathcal{D}_{\text{imgP}}}$ and $\Theta_{\mathcal{D}_{\text{textP}}}$, real trajectory matching length $M$, synthetic trajectory matching length $T$; total optimization iteration number $Iter$

1: Initialize $\mathcal{S}$ with $|\mathcal{S}|$ randomly sampled image-text pairs and one-hot groundtruth labels
2: Load pretrained weights into encoders (frozen); randomly initialize projection heads
3: **for** $it = 1$ to $Iter$ **do**
4:      Sample $\theta^t_{\mathcal{D}_{\text{imgP}}}$, $\theta^t_{\mathcal{D}_{\text{textP}}}$ and $\theta^{t+M}_{\mathcal{D}_{\text{imgP}}}$, $\theta^{t+M}_{\mathcal{D}_{\text{textP}}}$ from $\Theta_{\mathcal{D}_{\text{imgP}}}$ and $\Theta_{\mathcal{D}_{\text{textP}}}$
5:      Initialize $\theta^t_{\mathcal{S}_{\text{imgP}}}$ and $\theta^t_{\mathcal{S}_{\text{textP}}}$ using $\theta^t_{\mathcal{D}_{\text{imgP}}}$ and $\theta^t_{\mathcal{D}_{\text{textP}}}$
6:      **for** $i = 1$ to $T$ **do**
7:          **for** mini-batch $\mathcal{B} = \{(\tilde{\boldsymbol{x}}_b, \tilde{\boldsymbol{\tau}}_b), \tilde{\boldsymbol{y}}_b\}^{|\mathcal{B}|}_{b=1} \in \mathcal{S}$ **do**
8:              Calculate image representaion $\{f^{\text{imgE}}(\tilde{\boldsymbol{x}}_b)\}$
9:              ▷ `Blending in representation space`
10:              $\{f^{\text{imgE}}(\tilde{\boldsymbol{x}}_b), \tilde{\boldsymbol{\tau}}_b\} = \text{RepBlend}(\{f^{\text{imgE}}(\tilde{\boldsymbol{x}}_b), \tilde{\boldsymbol{\tau}}_b\})$
11:              Compute loss $\mathcal{L}^{\mathcal{B}}_{\text{wBCE}}$ using Equation 2
12:              Update projection head weights $\theta^{t+i}_{\mathcal{S}_{\text{imgP}}}$ and $\theta^{t+i}_{\mathcal{S}_{\text{textP}}}$
13:          **end for**
14:          ▷ `Symmetric projection trajectory matching`
15:          Optimize $\mathcal{S} = \{(\tilde{\boldsymbol{x}}_j, \tilde{\boldsymbol{\tau}}_j), \tilde{\boldsymbol{y}}_j\}^{|\mathcal{S}|}_{j=1}$ according to Equation 6
16:      **end for**
17: **end for**
**Ensure:** Synthetic dataset $\mathcal{S}$

---

**Baselines.** The comparison encompasses a range of SOTA approaches, including coreset selection methods such as Random sampling, Herding [52], K-Center [16], and Forgetting [47], as well as recent advances in dataset distillation tailored for vision-language models, including MTT-VL [53], TESLA-VL [55], and LoRS [55]. A detailed description of these methods can be found in the Appendix E. For fairness, both LoRS [55] and our method synthesize one fewer pair per distillation budget (e.g., 99 pairs for a budget of 100) to account for the additional similarity-matrix overhead.

**Implementation Details.** We construct a CLIP-style architecture using the aforementioned image and text encoders. The image encoder is initialized with ImageNet-pretrained weights [8], while the text encoder is initialized with the official pretrained weights provided by the corresponding language model. After feature extraction, the outputs from both branches are passed through separate linear projection layers to obtain the final embeddings. During buffer generation, distillation, and evaluation training, the encoders are frozen and only the projection layers are optimized. We collect 20 expert trajectories, each consisting of 10 training epochs. The hyperparameter settings follow those used in LoRS [55] and can be found in Table 7 and Table 8 in Appendix F.

### 4.2 Main Results

The results on Flickr-30K [36] and MS-COCO [27] are presented in Table 1 and Table 2, respectively. Our method consistently outperforms all baseline methods, across all distillation budgets and evaluation metrics. Notably, on Flickr-30k, under the extremely low-data regime of 100 training pairs (0.3%), our method achieves an IR@1 of 11.5%, substantially surpassing LoRS (8.3%) and MTT-VL (4.7%). Similarly, our TR@10 reaches 55.5%, a considerable gain over the best baseline LoRS (49.2%). These trends hold consistently across all pair settings. Under the 500-pair scenario (1.7%), our method improves the IR@10 from 41.6% (LoRS) to 55.9% and TR@10 from 53.7% to 66.7%, reflecting a relative gain of over 30%. On MS-COCO, a dataset known for higher complexity and variability, our method continues to exhibit superior performance. Under the 100-pair setting (0.8‰), our approach achieves IR@10 = 22.3% and TR@10 = 28.0%, substantially outperforming LoRS, which attains 12.2% and 19.6%, respectively. At a higher budget of 500 training pairs (4.4‰), our method maintains its advantage, achieving the highest IR@10 (30.6%) and TR@10 (32.9%) among all evaluated methods. Besides, we also extend our method to a larger-scale setting using the LLaVA-cc3m dataset, which serves as the pretraining dataset for LLaVA and consists of 558k image-text pairs. We use approximately 60% of the data (about 334k pairs) for training and reserve a

Table 1: Results on Flickr-30k [36]. Both distillation and validation are performed using NFNet+BERT. The model trained on full dataset performs: IR@1=23.16, IR@5=53.98, IR@10=66.62; TR@1=33.8, TR@5=65.7, TR@10=76.9.

| Pairs | Ratio | Metric | Coreset Selection | | | | Dataset Distillation | | | |
|---|---|---|---|---|---|---|---|---|---|---|
| | | | Rand | Herd [52] | K-Cent [16] | Forget [47] | MTT-VL [53] | TESLA-VL [55] | LoRS [55] | Ours |
| 100 | 0.3% | IR@1 | 1.0 | 0.7 | 0.7 | 0.7 | $4.7_{\pm0.2}$ | $0.5_{\pm0.2}$ | $8.3_{\pm0.2}$ | $\mathbf{11.5}_{\pm0.4}$ |
| | | IR@5 | 4.0 | 2.8 | 3.1 | 2.4 | $15.7_{\pm0.5}$ | $2.3_{\pm0.2}$ | $24.1_{\pm0.2}$ | $\mathbf{32.0}_{\pm0.7}$ |
| | | IR@10 | 6.5 | 5.3 | 6.1 | 5.6 | $24.6_{\pm1.0}$ | $4.7_{\pm0.4}$ | $35.1_{\pm0.3}$ | $\mathbf{44.5}_{\pm0.6}$ |
| | | TR@1 | 1.3 | 1.1 | 0.6 | 1.2 | $9.9_{\pm0.3}$ | $5.5_{\pm0.5}$ | $11.8_{\pm0.2}$ | $\mathbf{16.2}_{\pm0.8}$ |
| | | TR@5 | 5.9 | 4.7 | 5.0 | 4.2 | $28.3_{\pm0.5}$ | $19.5_{\pm0.9}$ | $35.8_{\pm0.6}$ | $\mathbf{41.7}_{\pm0.9}$ |
| | | TR@10 | 10.1 | 7.9 | 7.6 | 9.7 | $39.1_{\pm0.7}$ | $28.9_{\pm1.0}$ | $49.2_{\pm0.5}$ | $\mathbf{55.5}_{\pm0.4}$ |
| 200 | 0.7% | IR@1 | 1.1 | 1.5 | 1.5 | 1.2 | $4.6_{\pm0.9}$ | $0.2_{\pm0.1}$ | $8.6_{\pm0.3}$ | $\mathbf{12.7}_{\pm0.8}$ |
| | | IR@5 | 4.8 | 5.5 | 5.4 | 3.1 | $16.0_{\pm1.6}$ | $1.3_{\pm0.2}$ | $25.3_{\pm0.2}$ | $\mathbf{34.7}_{\pm0.6}$ |
| | | IR@10 | 9.2 | 9.3 | 9.9 | 8.4 | $25.5_{\pm2.6}$ | $2.5_{\pm0.2}$ | $36.6_{\pm0.3}$ | $\mathbf{47.6}_{\pm0.5}$ |
| | | TR@1 | 2.1 | 2.3 | 2.2 | 1.5 | $10.2_{\pm0.8}$ | $2.8_{\pm0.5}$ | $14.5_{\pm0.5}$ | $\mathbf{18.6}_{\pm0.7}$ |
| | | TR@5 | 8.7 | 8.4 | 8.2 | 8.4 | $28.7_{\pm1.0}$ | $10.4_{\pm1.5}$ | $38.7_{\pm0.5}$ | $\mathbf{46.0}_{\pm0.8}$ |
| | | TR@10 | 13.2 | 14.4 | 13.5 | 10.2 | $41.9_{\pm1.9}$ | $17.4_{\pm1.6}$ | $53.4_{\pm0.5}$ | $\mathbf{60.0}_{\pm0.6}$ |
| 500 | 1.7% | IR@1 | 2.4 | 3.0 | 3.5 | 1.8 | $6.6_{\pm0.3}$ | $1.1_{\pm0.2}$ | $10.0_{\pm0.2}$ | $\mathbf{17.0}_{\pm0.6}$ |
| | | IR@5 | 10.5 | 10.0 | 10.4 | 9.0 | $20.2_{\pm1.2}$ | $7.3_{\pm0.4}$ | $28.9_{\pm0.7}$ | $\mathbf{42.5}_{\pm0.5}$ |
| | | IR@10 | 17.4 | 17.0 | 17.3 | 15.9 | $30.0_{\pm2.1}$ | $12.6_{\pm0.5}$ | $41.6_{\pm0.6}$ | $\mathbf{55.9}_{\pm0.6}$ |
| | | TR@1 | 5.2 | 5.1 | 4.9 | 3.6 | $13.3_{\pm0.6}$ | $5.1_{\pm0.2}$ | $15.5_{\pm0.7}$ | $\mathbf{22.5}_{\pm0.4}$ |
| | | TR@5 | 18.3 | 16.4 | 16.4 | 12.3 | $32.8_{\pm1.8}$ | $15.3_{\pm0.5}$ | $39.8_{\pm0.4}$ | $\mathbf{53.2}_{\pm0.3}$ |
| | | TR@10 | 25.7 | 24.3 | 23.3 | 19.3 | $46.8_{\pm0.8}$ | $23.8_{\pm0.3}$ | $53.7_{\pm0.3}$ | $\mathbf{66.7}_{\pm0.3}$ |

Table 2: Results on MS-COCO [27]. Both distillation and validation are performed using NFNet+BERT. The model trained on full dataset performs: IR@1=14.6, IR@5=38.9, IR@10=53.2; TR@1=20.6, TR@5=46.8, TR@10=61.3.

| Pairs | Ratio | Metric | Coreset Selection | | | | Dataset Distillation | | | |
|---|---|---|---|---|---|---|---|---|---|---|
| | | | Rand | Herd [52] | K-Cent [16] | Forget [47] | MTT-VL [53] | TESLA-VL [55] | LoRS [55] | Ours |
| 100 | 0.8‰ | IR@1 | 0.3 | 0.5 | 0.4 | 0.3 | $1.3_{\pm0.1}$ | $0.3_{\pm0.2}$ | $1.8_{\pm0.1}$ | $\mathbf{4.1}_{\pm0.3}$ |
| | | IR@5 | 1.3 | 1.4 | 1.4 | 1.5 | $5.4_{\pm0.3}$ | $1.0_{\pm0.4}$ | $7.1_{\pm0.2}$ | $\mathbf{13.9}_{\pm0.8}$ |
| | | IR@10 | 2.7 | 3.5 | 2.5 | 2.5 | $9.5_{\pm0.5}$ | $1.8_{\pm0.5}$ | $12.2_{\pm0.2}$ | $\mathbf{22.3}_{\pm0.5}$ |
| | | TR@1 | 0.8 | 0.8 | 1.4 | 0.7 | $2.5_{\pm0.3}$ | $2.0_{\pm0.2}$ | $3.3_{\pm0.2}$ | $\mathbf{5.2}_{\pm0.5}$ |
| | | TR@5 | 3.0 | 2.1 | 3.7 | 2.6 | $10.0_{\pm0.5}$ | $7.7_{\pm0.5}$ | $12.2_{\pm0.3}$ | $\mathbf{17.9}_{\pm0.9}$ |
| | | TR@10 | 5.0 | 4.9 | 5.5 | 4.8 | $15.7_{\pm0.4}$ | $13.5_{\pm0.3}$ | $19.6_{\pm0.3}$ | $\mathbf{28.0}_{\pm0.3}$ |
| 200 | 1.7‰ | IR@1 | 0.6 | 0.9 | 0.7 | 0.6 | $1.7_{\pm0.1}$ | $0.1_{\pm0.1}$ | $2.4_{\pm0.1}$ | $\mathbf{6.1}_{\pm0.8}$ |
| | | IR@5 | 2.3 | 2.4 | 2.1 | 2.8 | $6.5_{\pm0.4}$ | $0.2_{\pm0.1}$ | $9.3_{\pm0.2}$ | $\mathbf{19.3}_{\pm0.7}$ |
| | | IR@10 | 4.4 | 4.1 | 5.8 | 4.9 | $12.3_{\pm0.8}$ | $0.5_{\pm0.1}$ | $15.5_{\pm0.2}$ | $\mathbf{29.8}_{\pm0.5}$ |
| | | TR@1 | 1.0 | 1.0 | 1.2 | 1.1 | $3.3_{\pm0.2}$ | $0.7_{\pm0.2}$ | $4.3_{\pm0.1}$ | $\mathbf{6.9}_{\pm0.6}$ |
| | | TR@5 | 4.0 | 3.6 | 3.8 | 3.5 | $11.9_{\pm0.6}$ | $3.1_{\pm0.5}$ | $14.2_{\pm0.3}$ | $\mathbf{21.8}_{\pm0.9}$ |
| | | TR@10 | 7.2 | 7.7 | 7.5 | 7.0 | $19.4_{\pm1.2}$ | $5.3_{\pm0.8}$ | $22.6_{\pm0.2}$ | $\mathbf{32.3}_{\pm0.7}$ |
| 500 | 4.4‰ | IR@1 | 1.1 | 1.7 | 1.1 | 0.8 | $2.5_{\pm0.5}$ | $0.8_{\pm0.2}$ | $2.8_{\pm0.2}$ | $\mathbf{6.2}_{\pm0.1}$ |
| | | IR@5 | 5.0 | 5.3 | 6.3 | 5.8 | $8.9_{\pm0.7}$ | $3.6_{\pm0.6}$ | $9.9_{\pm0.5}$ | $\mathbf{19.9}_{\pm0.3}$ |
| | | IR@10 | 8.7 | 9.9 | 10.5 | 8.2 | $15.8_{\pm1.5}$ | $6.7_{\pm0.9}$ | $16.5_{\pm0.7}$ | $\mathbf{30.6}_{\pm0.1}$ |
| | | TR@1 | 1.9 | 1.9 | 2.5 | 2.1 | $5.0_{\pm0.4}$ | $1.7_{\pm0.4}$ | $5.3_{\pm0.5}$ | $\mathbf{7.0}_{\pm0.2}$ |
| | | TR@5 | 7.5 | 7.8 | 8.7 | 8.2 | $17.2_{\pm1.3}$ | $5.9_{\pm0.8}$ | $18.3_{\pm1.5}$ | $\mathbf{22.0}_{\pm0.3}$ |
| | | TR@10 | 12.5 | 13.7 | 14.3 | 13.0 | $26.0_{\pm1.9}$ | $10.2_{\pm1.0}$ | $27.9_{\pm1.4}$ | $\mathbf{32.9}_{\pm0.6}$ |

non-overlapping set of 10k pairs for validation[4]. In addition, we evaluate our approach with more powerful encoders, including DiNo-v2 [34] (85,798,656 parameters) for vision and BGE-1.5 [54] (109,482,240 parameters) for text. The results (shown in Table 3) demonstrate that our method remains effective when scaling both model capacity and training data, and it significantly outperforms the SOTA competitor. Moreover, our method also demonstrates strong generalizability to other multimodal settings, such as audio-text benchmark. See Appendix H for details.

### 4.3 Ablation Study

**Representation Blending & Symmetric Matching.** We conduct an ablation study on the Flickr-30K dataset using NFNet+BERT to evaluate the individual and combined contributions of the proposed components: Representation Blending (RB) and Symmetric Projection Trajectory Matching (SM). As shown in Figure 5, removing either module leads to consistent performance degradation across all retrieval metrics (IR@1/5/10 and TR@1/5/10) and distillation budgets (100, 200, 500 pairs). RB contributes by mitigating intra-modal collapse; as illustrated in Figure 3, it effectively reduces

---

[4]see `https://huggingface.co/xinxin66/RepBlend/tree/main/datasets/cc3m`.

Table 3: Results when scaling to larger dataset and model. NFNet + BERT on LLaVA-cc3m: the model trained on the full dataset performs: IR@1=9.13, IR@5=25.94, IR@10=36.34, TR@1=9.49, TR@5=26.08, TR@10=37.07. DiNo-v2 + BGE-1.5 on MS-COCO: the model trained on the full dataset performs: IR@1=22.70, IR@5=51.13, IR@10=65.26, TR@1=31.04, TR@5=61.96, TR@10=74.1.

| Pairs | Methods | NFNet + BERT on LLaVA-cc3m | | | | | | DiNo-v2 + BGE-1.5 on MS-COCO | | | | | |
| | | IR@1 | IR@5 | IR@10 | TR@1 | TR@5 | TR@10 | IR@1 | IR@5 | IR@10 | TR@1 | TR@5 | TR@10 |
|---|---|---|---|---|---|---|---|---|---|---|---|---|---|
| 200 | LoRS [55] | 1.56 | 5.33 | 8.77 | 1.01 | 4.04 | 6.81 | 1.59 | 6.12 | 10.59 | 1.62 | 6.15 | 10.47 |
| | Ours | **3.53** | **12.38** | **19.33** | **4.39** | **13.45** | **20.18** | **12.52** | **31.96** | **44.39** | **16.06** | **36.84** | **49.34** |
| 500 | LoRS [55] | 1.96 | 6.72 | 10.55 | 1.41 | 5.11 | 8.51 | 2.48 | 8.46 | 13.42 | 3.18 | 10.36 | 16.37 |
| | Ours | **4.45** | **15.09** | **22.31** | **5.14** | **14.89** | **22.97** | **13.37** | **33.09** | **45.69** | **16.90** | **39.38** | **52.14** |
| 800 | LoRS [55] | 1.67 | 5.87 | 9.69 | 1.68 | 6.11 | 10.25 | 2.95 | 9.90 | 15.69 | 4.56 | 13.66 | 20.63 |
| | Ours | **5.26** | **16.31** | **24.23** | **5.42** | **16.13** | **24.16** | **13.68** | **33.58** | **45.93** | **17.14** | **39.76** | **52.92** |

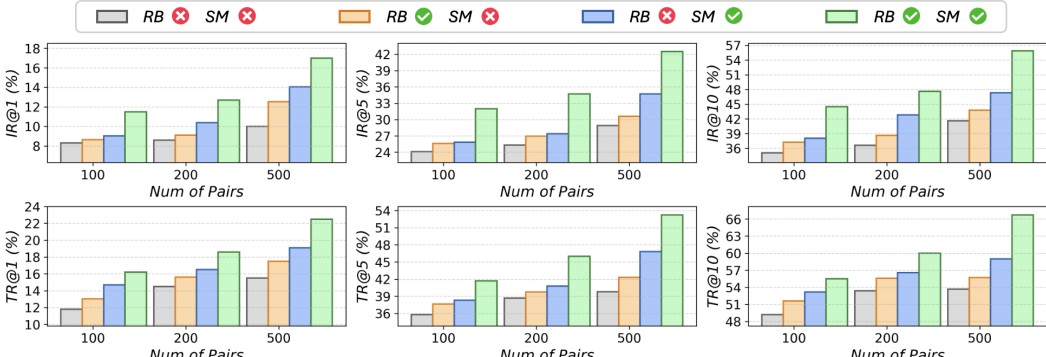

Figure 5: Ablation study of Representation Blending (RB) and Symmetric Projection Trajectory Matching (SM) on Flickr-30K with NFNet+BERT.

intra-modal similarity and enhances representational diversity. SM further balances the learning dynamics across modalities and improves cross-modal alignment, as evidenced in Figure 4. When combined, RB and SM achieve the best overall performance, highlighting their complementary roles in enhancing intra-modal diversity and cross-modal alignment.

**Cross-Architecture Generalization.** We further validate the generalization capability of RepBlend across diverse architectures. Following the protocol of LoRS [55], we keep the text encoder fixed and evaluate the dataset distilled with NFNet+BERT using alternative image encoders, including ResNet-50 and RegNet. As shown in Table 4, RepBlend consistently maintains strong performance across different encoder architectures. Moreover, we extend the evaluation to a broader set of architecture combinations, such as ResNet-50+BERT, ViT+BERT, RegNet+BERT, and NFNet+DistilBERT, as illustrated in Figure 6 and Figure 7 in Appendix I. Across all architectures, datasets, and distillation budgets, RepBlend consistently outperforms the sota baseline, demonstrating its robustness and architectural adaptability.

**Zero-Shot Generalization.** To further validate the effectiveness of our distilled dataset, we further evaluate zero-shot ImageNet [8] classification and OCR-relevant retrieval on TextCaps [45]. Specifically, we randomly select 10 classes from ImageNet-1K and report Top-1 and Top-5 zero-shot accuracies. For TextCaps, we measure retrieval performance on 3,166 validation samples. The results, summarized in Table 5, show that under the same budget, models trained on our distilled dataset outperform LoRS and narrow the performance gap to the full-dataset baseline.

## 4.4 Computational Efficiency

In the proposed method, the training trajectories of image and text projection layers are used for matching optimization. Although we introduce an additional image projection, it incurs negligible computational overhead. In fact, as shown in Table 6, our method achieves significantly better computational efficiency compared to prior work. Specifically, the time required to construct expert trajectories is reduced from 70 minutes to 40 minutes per trajectory (1.75× speedup), and the corresponding memory footprint decreases from 1.63 GB to 0.73 GB (2.23× reduction). During the distillation phase, our method accelerates training iterations from 11.5 seconds to 1.71 seconds per iteration, yielding a 6.7× speedup. Moreover, it lowers the peak GPU memory usage from 21.78 GB

Table 4: Cross-architecture generalization. The distilled data are synthesized with NFNet+BERT and evaluated across architectures on Flickr-30K under the 500-pair setting.

| Evaluate Model | Methods | IR@1 | IR@5 | IR@10 | TR@1 | TR@5 | TR@10 |
|---|---|---|---|---|---|---|---|
| ResNet+BERT | TESLA-VL [55] | $3.0_{\pm 0.2}$ | $10.8_{\pm 0.5}$ | $17.0_{\pm 0.8}$ | $6.0_{\pm 0.9}$ | $18.8_{\pm 0.7}$ | $27.7_{\pm 1.2}$ |
| | LoRS [55] | $3.3_{\pm 0.2}$ | $12.7_{\pm 0.3}$ | $20.4_{\pm 0.2}$ | $6.8_{\pm 0.2}$ | $19.6_{\pm 1.3}$ | $31.1_{\pm 0.3}$ |
| | Ours | $\mathbf{4.2}_{\pm 0.2}$ | $\mathbf{14.1}_{\pm 0.2}$ | $\mathbf{23.6}_{\pm 0.6}$ | $\mathbf{8.4}_{\pm 0.2}$ | $\mathbf{23.1}_{\pm 0.8}$ | $\mathbf{35.0}_{\pm 1.3}$ |
| RegNet+BERT | TESLA-VL [55] | $3.2_{\pm 0.8}$ | $11.1_{\pm 1.8}$ | $17.5_{\pm 1.3}$ | $5.8_{\pm 0.1}$ | $18.6_{\pm 0.6}$ | $28.1_{\pm 1.0}$ |
| | LoRS [55] | $3.5_{\pm 0.1}$ | $12.6_{\pm 0.3}$ | $21.1_{\pm 0.4}$ | $6.8_{\pm 0.3}$ | $20.8_{\pm 0.3}$ | $30.2_{\pm 0.3}$ |
| | Ours | $\mathbf{3.9}_{\pm 0.2}$ | $\mathbf{13.9}_{\pm 0.3}$ | $\mathbf{24.0}_{\pm 0.6}$ | $\mathbf{7.9}_{\pm 0.3}$ | $\mathbf{24.2}_{\pm 0.3}$ | $\mathbf{36.2}_{\pm 1.1}$ |

Table 5: Zero-Shot Generalization. Models trained on the distilled MS-COCO dataset under the 500-pair setting are evaluated on zero-shot ImageNet classification and TextCaps retrieval tasks.

| Methods | ImageNet-10 Classification | | TextCaps Retrieval | | | | | |
|---|---|---|---|---|---|---|---|---|
| | ACC@1 | ACC@5 | IR@1 | IR@5 | IR@10 | TR@1 | TR@5 | TR@10 |
| LoRS [55] | 21.4 | 74.4 | 1.7 | 5.1 | 8.4 | 0.4 | 1.7 | 3.1 |
| Ours | **27.6** | **76.2** | **3.1** | **9.4** | **14.5** | **1.9** | **6.2** | **10.3** |

to 10.17 GB (2.14× reduction). These results show that our projection-based design not only enables effective multimodal distillation, but also leads to substantially improved computational efficiency.

# 5 Conclusion

In this work, we investigate the underexplored challenge of modality collapse in multimodal dataset distillation (MDD), where intra-modal similarity is excessively amplified and inter-modal alignment is degraded. Through theoretical analysis and empirical evidence, we attribute this phenomenon to the inherent over-compression behavior of dataset distillation and its interplay with cross-modal contrastive supervision. To mitigate these issues, we propose RepBlend, a novel MDD framework incorporating two key components: Representation Blending for enhancing intra-modal diversity and Symmetric Projection Trajectory Matching for achieving balanced and effective supervision across modalities. Extensive experiments on Flickr-30K and MS-COCO confirm the superiority of RepBlend in both retrieval performance and distillation efficiency.

Table 6: Study of computational efficiency.

| Methods | LoRS [55] | Ours |
|---|---|---|
| (IR@1, TR@1) (%) | (8.3, 11.8) | **(11.5, 16.2)** |
| Buffer | | |
| Speed (min/traj) | 70 | **40** |
| Memory (GB/traj) | 1.63 | **0.73** |
| Distillation | | |
| Speed (s/iter) | 11.5 | **1.71** |
| Peak GPU VRAM (GB) | 21.78 | **10.17** |

**Limitations and Future work.** Despite the promising results of RepBlend, current MDD frameworks, including ours, remain limited to pair-level modeling, which restricts fine-grained alignment between text tokens and visual objects. Additionally, insufficient cross-instance interaction hampers representation expressiveness and limits further gains in compression. In the future, we will explore instance-aware, relation-enhanced strategies to overcome these challenges.

# 6 Acknowledgement

This research is supported by Xin Zhang's A*STAR Career Development Fund (CDF) (Project No. C243512009), and Jiawei Du's A*STAR Career Development Fund (CDF) (Project No. C233312004). This research is also supported by the Japan Science and Technology Agency (JST) and the Agency for Science, Technology and Research (A*STAR) under the Japan-Singapore Joint Call (Project No. R24I6IR133).

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
