# OpenReview forum: "Beyond Modality Collapse: Representation Blending for Multimodal Dataset Distillation"
_NeurIPS.cc/2025/Conference — NeurIPS 2025 poster_

### Official Review · Reviewer_nvV9 · 2025-07-01

**Clarity:** 3
**Significance:** 2
**Originality:** 2
**Rating:** 4
**Confidence:** 3

**Summary:**

The paper first identifies the "modality collapse" in multimodal dataset distillation and then provides theoretical and empirical analysis on the cause. To mitigate, the authors propose RepBlend to enhance intra-modal diversity and inter-modal similarity. And lastly they have the appraoch tested on a bunch of datasets including Flickr-30K and MS-COCO, showing the best performance among the compared methods.

**Questions:**

Q1. To reach the full performance, roughly how many samples are needed? How does this number vary across different methods? Hope to see the plot with more numbers.

Q2. How's the performance with stronger base models, like DinoV2, or BGE, NV-Embed?

Q3. How's the zero-shot performance, how well can the trained model generalize to other datasets?

**Ethical Concerns:**

["NO or VERY MINOR ethics concerns only"]

**Final Justification:**

I'll maintain my score of 4. The author's response resolved my questions on the metric correlation, zero-shot and unimodal performance. But I'm still questioning the originality of the idea (similar to FuseMix). However, the proposed method significantly outperforms the listed baselines which gives me more confidence in accepting the paper. After revisiting all these aspects, I'm more comfortable with maintaining my score.

**Limitations:**

See weaknesses and questions

**Quality:**

3

**Strengths And Weaknesses:**

**Strengths**

S1. The authors identified the "modality gap" issue in the realm of multimodal dataset distillation, and attributed the issue to the over-compression behavior through analysis. Motivation is intuitive and reasonable.

S2. The multi-modal results presented in the paper demonstrate a noticeable improvement over previous methods

S3. Overall writing is good, easy to follow.

**Weaknesses**

W1. Similar approaches have already been introduced in prior work FuseMix [C1] and should be discussed.

W2. The authors employed the intra-modal similarity and modality gap metrics but didn't not discuss this in detail, e.g., how will these individually affect the performance.

W3. The experiments are focused on multi-modal performance only. However, some experiments on unimodal task will strengthen the paper. To support the claim that the approach reduces both intra-modal similarity and modality gap, showing evidence that this improves not only multi-modal but also uni-modal performance is helpful.

C1. Noël Vouitsis, Zhaoyan Liu, Satya Krishna Gorti, Valentin Villecroze, Jesse C. Cresswell, Guangwei Yu, Gabriel
Loaiza-Ganem, and Maksims Volkovs. Data-efficient multimodal fusion on a single gpu. , CVPR 2024

---

> ### Author Rebuttal · Authors · 2025-07-31
>
> **Q1 (Weaknesses 1):** Similar approaches have already been introduced in prior work FuseMix [1] and should be discussed.
>
> **A1:** Thank you for mentioning this important prior work. FuseMix is a latent data augmentation approach proposed to facilitate efficient multimodal alignment. The distinction between FuseMix and our method lies in both motivation and mechanism. In the final version, we will include a dedicated section for this discussion.
>
> - **Motivation**:
> FuseMix assumes that well‑pretrained unimodal encoders already capture rich semantics, leveraging them for multimodal alignment without expensive full‑model training. It employs latent data augmentation to mitigate data scarcity and the limited capacity of lightweight adapters. In contrast, our method targets the practical challenge of modality collapse in multimodal dataset distillation. Through theoretical analysis, we show that Representation Blending alleviates this issue, enhancing intra‑modal diversity and cross‑modal alignment.
>
> - **Mechanism**:
> FuseMix is essentially a data augmentation method, where even modality‑agnostic perturbations such as Gaussian noise can also boost baseline performance. In contrast, our approach directly addresses modality collapse. As shown in Section 3.2, Gaussian noise introduces semantically meaningless signals that enlarge the modality gap and degrade cross‑modal alignment, leading to a performance drop. These results underscore the distinctiveness of our design for multimodal dataset distillation (MDD).
>
> ---
> **Q2 (Weaknesses 2):** The authors used intra-modal similarity and modality gap but did not detail their individual impact on performance.
>
> **A2:** Thank you for the question. The intra-modal similarity (Sim), concentration ratio (CR), and modality gap (Gap) are key metrics for evaluating the quality of the distilled dataset. In general, smaller Sim and CR indicate greater intra-modal diversity, while a smaller Gap reflects stronger cross-modal alignment. Together, these factors lead to better retrieval performance. However, as these metrics are interdependent, it is difficult to vary one without affecting the others. So here we provide two cases to generally assess their individual effects.
>
> - *Distilling 100 pairs on Flickr-30k using NFNet+BERT.*
>
> || Sim (CR) | Gap | IR@1         | IR@5         | IR@10        | TR@1         | TR@5         | TR@10
> |-------- | -------- | -------- | -------- |-------- |-------- |-------- |-------- |-------- |
> |baseline| 0.522 (0.95)    | 0.327     | 8.30 | 24.10 | 35.10 | 11.80 | 35.80 | 49.20
> |case 1| 0.511 (0.54)    | 0.318     |8.64|25.64|37.24|13.03|37.63|51.6
> |case 2| 0.512 (0.57)    | 0.23     |11.50 | 32.00 | 44.50 | 16.20 | 41.70 | 55.50
>
> - Baseline vs. Case 1: A substantial reduction in intra-modal similarity (CR from 0.95 → 0.54) led to a 2.1% increase in IR@10 and a 2.4% increase in TR@10.
>
> - Case 1 vs. Case 2: A further notable reduction in the cross-modal gap (0.318 → 0.23) yielded an additional 7.3% increase in IR@10 and 3.9% in TR@10 compared with Case 1.
>
> These results indicate that reducing intra-modal concentration, and narrowing the modality gap, provides complementary benefits and leads to substantial performance gains.
>
> ---
> **Q3 (Weaknesses 3):** Some experiments on unimodal task will strengthen the paper.
>
> **A3:** Thank you for the insightful suggestion. We evaluated the proposed Representation Blending (RB) in the unimodal setting by adopting SRe2L [2] as the baseline and integrating RB into its distillation process. RB effectively reduces feature redundancy and promotes more class-discriminative representations. As shown below, RB consistently improves SRe2L’s performance across various IPC budgets.
>
> - *Distill CIFAR-100 with ResNet-18*
>
> | IPC | SRe2L [2] | SRe2L + RB |
> | -------- | -------- | -------- |
> | 1     | 7.3     | **7.7**    |
> | 10     | 26.3     | **27.4**     |
> | 50     |  51.6    |  **52.8**    |
>
> ---
> **Q4 (Questions 1):** To reach the full performance, roughly how many samples are needed? How does this number vary across different methods? Hope to see the plot with more numbers.
>
> **A4:** We extended the comparison to larger pair settings and provide a detailed table since figures are not allowed in the rebuttal. Both LoRS and our method show only marginal gains as pairs increase, reflecting the well-known saturation issue in dataset distillation where added pairs bring limited diversity. Nevertheless, our method maintains consistent superiority over LoRS under the same conditions. In the future, we will further explore ways to overcome this saturation toward lossless or nearly lossless MDD.
>
> - *NFNet + Bert on Fliker-30k dataset. The model trained on the full dataset performs: IR@1=23.16, IR@5=53.98, IR@10=66.62; TR@1=33.8, TR@5=65.7, TR@10=76.9.*
>
> | pairs |Methods| IR@1 | IR@5 | IR@10| TR@1 | TR@5 | TR@10
> | -------- | -------- | -------- | -------- | -------- | -------- | -------- |-------- |
> |  199  |LoRS|  8.6    | 25.3     |36.6|14.5|38.7|53.4
> ||**Ours**|**12.7**   |  **34.7**    |**47.6**|**18.6**|**46.0**|**60.0**
> |  299  |LoRS|   9.6 |   27.7   | 39.3|13.82|39.32|53.5
> ||**Ours**|**14.3**  |   **36.1**   |**50.6**|**20.6**|**49.2**|**62.3**
> |  499  |LoRS |  10.0    |  28.9    |41.6|15.5|39.8|53.7
> ||**Ours**|**17.0**    |   **42.5**   |**55.9**|**22.5**|**53.2**|**66.7**
> |  699  | LoRS|  11.6   |    31.5  |44.1|16.3|41.9|57.0
> ||**Ours**|**17.4**  |   **43.5**   |**56.8**|**22.8**|**53.4**|**67.1**
> |  799  | LoRS|  11.5   |   31.5   |44.1|15.4|40.6|56.8
> ||**Ours**|**17.6**      |**44.0**      |**56.9**|**22.2**|**53.7**|**67.0**
>
> - *NFNet + Bert on COCO dataset. The model trained on the full dataset performs: IR@1=14.6, IR@5=38.9, IR@10=53.2; TR@1=20.6, TR@5=46.8, TR@10=61.3.*
>
> |pairs| Methods | IR@1 | IR@5 | IR@10| TR@1 | TR@5 | TR@10
> | -------- | -------- | -------- | -------- | -------- | -------- | -------- | -------- |
> |  199  |LoRS| 2.4     |  9.3    |15.5|4.3|14.2|22.6
> ||**Ours**|  **6.1**    |   **19.3**   |**29.8**|**6.9**|**21.8**|**32.3**
> |  299  |LoRS|   2.5   |   9.5 |15.9|4.4|15.8|25.1
> ||**Ours**|   **6.6**  |   **20.5**   |**29.5**|**6.9**|**21.7**|**32.2**
> |  499  |LoRS|  2.8    |   9.9   |16.5|5.3|18.3|27.9
> ||**Ours**| **6.2**    |  **19.9**    |**30.6**|**7.0**|**22.0**|**32.9**
> |  699  |LoRS|  3.0    |   10.9   |17.7|6.0|19.0|28.1
> ||**Ours**| **6.5**     |    **20.1**  |**31.0**|**7.5**|**22.2**|**33.4**
> |  799  |LoRS|3.1|10.7|17.4|5.9|19.1|28.5
> ||**Ours**| **6.7**    |     **20.8** |**31.7**|**7.7**|**22.5**|**33.7**
>
> ---
> **Q5 (Questions 2):** How's the performance with stronger base models, like DinoV2, or BGE, NV-Embed?
>
> **A5:** Thank you for the question. We have evaluated our approach with more powerful encoders, like ***DiNo-v2*** (85,798,656 parameters) for vision and ***BGE-1.5*** (109,482,240 parameters) for text. The results demonstrate that our method remains effective when scaling model capacity.
>
> - *DiNo-v2 + BGE-1.5 on COCO dataset. The model trained on the full dataset performs: IR@1=22.70, IR@5=51.13, IR@10=65.26, TR@1=31.04, TR@5=61.96, TR@10=74.1.*
>
> | pairs| Methods | IR@1 | IR@5 | IR@10| TR@1 | TR@5 | TR@10
> | --------| -------- | -------- | -------- | -------- | -------- | -------- | -------- |
> | 199| LoRS     |1.59|6.12|10.59|1.62|6.15|10.47
> |      | **ours**     |  **12.52**    |**31.96**|**44.39**|**16.06**|**36.84**|**49.34**
> | 499| LoRS     |2.48|8.46|13.42|3.18|10.36|16.37
> |      | **ours**     |  **13.37**    |**33.09**|**45.69**|**16.90**|**39.38**|**52.14**
>
> *DiNo-v2: https://huggingface.co/timm/vit_base_patch16_224.dino*
>
> *BGE-1.5: https://huggingface.co/BAAI/bge-base-en-v1.5*
>
> ---
> **Q6 (Questions 3):** How's the zero-shot performance, how well can the trained model generalize to other datasets?
>
> **A6:** Thank you for raising this point. We further validated the proposed method with zero-shot ImageNet classification (10 randomly selected classes) and OCR-relevant retrieval on TextCaps. Compared with LoRS, our method trained on only 499 distilled pairs achieves higher accuracy and retrieval performance, thereby narrowing the gap with the full-data baseline.
>
> - *ImageNet-10 zero-shot classification*
>
> | Methods     |Acc1 | Acc5
> | -------- | -------- | -------- |
> | Baseline     |   36.4 | 83.2|
> | LoRS    |  21.4 | 74.4|
> | **Ours**    |  **27.6**  | **76.2**|
>
> - *TextCaps zero-shot retrieval*
>
> | Methods     |IR@1 | IR@5         | IR@10     |  TR@1  | TR@5         | TR@10
> | -------- | -------- | -------- |-------- | -------- |-------- | -------- |
> | Baseline     |   7.4|18.1  | 26.2   |5.1|14.7|21.3
> | LoRS    |    1.7  |  5.1   |8.4|0.4|1.7|3.1|
> | **Ours**    |   **3.1**   |   **9.4**   |**14.5**|**1.9**|**6.2**|**10.3**
>
> Beyond zero-shot performance, we also extend our method to the large-scale ***LLaVA-cc3m*** dataset, which contains 558k image-text pairs. We train on ~60% (334k pairs) and validate on a separate 10k-pair set. The results demonstrate that our method generalizes well to large-scale datasets.
>
> - *NFNet + Bert on LLaVA-cc3m dataset. The model trained on the full dataset performs: IR@1=9.13, IR@5=25.94, IR@10=36.34, TR@1=9.49, TR@5=26.08, TR@10=37.07.*
>
> | pairs | Methods | IR@1 | IR@5 | IR@10| TR@1 | TR@5 | TR@10
> | -------- | -------- | -------- | -------- | -------- | -------- | -------- | -------- |
> | 199 | LoRS     |   1.56|5.33|8.77|1.01|4.04|6.81
> | | **ours**     | **3.53**     | **12.38**    |**19.33**|**4.39**|**13.45**|**20.18**
> | 499 | LoRS     |1.96|6.72|10.55|1.41|5.11|8.51
> | | **ours**     |**4.45**|**15.09**|**22.31**|**5.14**|**14.89**|**22.97**
> | 799 | LoRS     |1.67   |  5.87    |9.69|1.68|6.11|10.25
> | | **ours** |**5.26**|**16.31**|**24.23**|**5.42**|**16.13**|**24.16**
>
> *[1] Squeeze, Recover and Relabel: Dataset Condensation at ImageNet Scale From A New Perspective. NeurIPS 2023.*
>
> *[2] TextCaps: a Dataset for Image Captioning with Reading Comprehension, ECCV 2020.*
>
> *LLaVA-cc3m: https://github.com/haotian-liu/LLaVA/blob/main/docs/Data.md*

---

> > ### Comment · Reviewer_nvV9 · 2025-08-06
> >
> > Thanks authors for the detailed response. These addressed most of my questions and please add the clarification and discussion.

---

> > > ### Author Response · Authors · 2025-08-07
> > > **Thank you very much for your kind follow-up.**
> > >
> > > Thank you very much for your kind follow-up and encouraging words. We’re glad to hear that most of your concerns have been addressed, and we truly appreciate your thoughtful suggestions. We will make sure to incorporate the clarifications and discussions into the revised version accordingly.

---

### Official Review · Reviewer_t2RZ · 2025-07-01

**Clarity:** 3
**Significance:** 3
**Originality:** 3
**Rating:** 5
**Confidence:** 4

**Summary:**

The authors examine the problem of Multimodal Dataset Distillation (MDD), where the goal is to generate a synthetic dataset distilled from a real one, that can be used to train a CLIP style model while using fewer data points. The authors identify issues with existing techniques for this goal, and through representation blending and improvements on the optimization process demonstrate improvements over existing methods.

**Questions:**

- I would be grateful if the authors could elaborate on the point I raised above regarding Appendix B, and whether there are any other underlying assumptions in the arguments made there.

- Also, I would be grateful if the authors could let me know whether they have tried Representation Blending in the unimodal setting (as it seems that it might be useful there as well).

**Ethical Concerns:**

["NO or VERY MINOR ethics concerns only"]

**Final Justification:**

As mentioned in my original review, I think this is a solid paper, and reading the authors' responses to both mine and other Reviewers' comments further solidifiy this belief (with my only concern regarding implicit assumptions for modality collapse having been addressed). As such, I am keeping my recommendation of "Accept" for this work.

**Limitations:**

The authors have adequately described the limitations and potential impact of their work.

**Paper Formatting Concerns:**

No major concerns.

**Quality:**

3

**Strengths And Weaknesses:**

Strengths

- The paper is clearly written, and the points made by the authors are mostly well supported.

- Both ideas presented by the authors are solid and novel in this topic as far as I am aware:

  - Representation blending is well motivated. The authors use an example where they blend the representations of the model with noise, and show that blending by itself alleviates the identified modality collapse issue. This is a good foundation for blending representations with other samples, which is clearly shown to improve the intra-modal similarity between samples in Figure 3.

  - Symmetric Projection Trajectory Matching is a simple idea at its core (using a projection head for the image side of the architecture, instead of just the image encoder). Nevertheless, this is shown to greatly improve the loss in the image encoder in Figure 4. This is important, as matching trajectories is key to the synthetic dataset having similar behavior to the real one, in both image and text domains.

- Experimental evaluation is also convincing. The authors use a similar set of experiments to state-of-the-art prior work (LoRS) and clearly improve upon that. This demonstrates the solid performance of the methodology presented in this paper.

Weaknesses

- The authors make a theoretical argument for modality collapse, with the analysis present in Appendix B. Their argument is that the way the synthetic dataset is updated, embeddings eventually become closer to each other, even across samples that are only slightly similar. However, some steps of the analysis imply some approximations (specifically, that the encoders are trained well enough for negative samples to have approximately orthogonal embeddings). I believe that the authors can improve this analysis by clearly stating hidden assumptions about the embeddings in Appendix B, and even expanding upon the visual analysis in Figure 2 (for example, by explicitly including a CR-Iteration plot, as CR is the most salient metric).

- In Table 3, the NFNet+BERT evaluation model case is excluded. For completeness, it should be added to this table as well (although it can be inferred from other tables).

- Some more minor points:

  - Tables 1 and 2 should float to the bottom of the page for readability.

  - Table 4 can also become horizontal instead of vertical, so that it can float to top after Table 3.

  - I would recommend highlighting (bolding / different color) the difference between Equations 5 and 6, since at first glance they are too similar (the difference being the use of image encoder or image projection head).

  - In line 168, “blends” -> “blend”.

---

> ### Author Rebuttal · Authors · 2025-07-31
>
> Thank you for your time and insightful feedback. We have provided detailed responses to all your questions below and hope they effectively resolve your concerns.
>
> **Q1 (Weaknesses 1 & Questions 1):** The analysis in Appendix B relies on implicit assumptions (e.g., negative sample embeddings being approximately orthogonal) that should be explicitly stated.
>
> **A1:** Thanks for the insightful suggestion. This deviation indeed relies on the assumption that both the image and text encoders are well-pretrained, such that negative pairs are approximately orthogonal in the high-dimensional embedding space. This assumption is consistent with our practical setup, where we employ ImageNet-1K–pretrained image encoders (e.g., NFNet) and BookCorpus + English Wikipedia–pretrained text encoders (e.g., BERT). The validity of this assumption has also been supported by several prior works [1, 2]. To further substantiate it, we empirically computed the relevant values, obtaining a cross-modal negative pair mean cosine similarity of 0.0003 and an intra-modal (image) mean cosine similarity of 0.004.  For clarity, this assumption is explicitly emphasized in the revised version.
>
> ---
>
> **Q2 (Weaknesses 1):** Suggests expanding the visual analysis, for instance by adding a CR‑Iteration plot, to strengthen the argument.
>
> **A2:** Thank you for the suggestion. We chose intra-modal similarity for illustration because it is more intuitive to understand compared with CR. The CR–Iteration plot shows the same trend as the current similarity–Iteration plot, since CR is positively correlated with intra-modal similarity; higher intra-modal similarity corresponds to higher CR, and in particular, CR provides a more numerically salient perspective. As figure updates are not permitted during rebuttal, we will incorporate this additional plot in the final version.
>
> ---
> **Q3 (Weaknesses 2):**
> In Table 3, the NFNet+BERT evaluation model case is excluded. For completeness, it should be added to this table as well (although it can be inferred from other tables).
>
> **A3:** Thanks for pointing this out. We have included the NFNet + BERT case in Table 3 to ensure completeness and consistency across the reported results.
>
> *Table 3. Cross-architecture generalization. The distilled data are synthesized using NFNet+BERT and evaluated across different architectures. Evaluations are conducted on Flickr-30K under the 500-pair setting. For fairness, both LoRS [3] and ours synthesize one fewer pair, e.g., 499 pairs.*
> | Evaluate Model  | Methods        | IR@1         | IR@5         | IR@10        | TR@1         | TR@5         | TR@10        |
> |-----------------|----------------|--------------|--------------|--------------|--------------|--------------|--------------|
> | NFNet+BERT | TESLA-VL      | 1.1±0.2 | 7.3±0.4 | 12.6±0.5 | 5.1±0.2 | 15.3±0.5 | 23.8±0.3
> |                 | LoRS          | 10.0±0.2 | 28.9±0.7 | 41.6±0.6 | 15.5±0.7 | 39.8±0.4 | 53.7±0.3
> |                 | **Ours**       | **17.0±0.6** | **42.5±0.5** | **55.9±0.6** | **22.5±0.4** | **53.2±0.3** | **66.7±0.3** |
> | ResNet+BERT | TESLA-VL       | 3.0±0.2    | 10.8±0.5   | 17.0±0.8   | 6.0±0.9    | 18.8±0.7   | 27.7±1.2   |
> |                 | LoRS           | 3.3±0.2    | 12.7±0.3   | 20.4±0.2   | 6.8±0.2    | 19.6±1.3   | 31.1±0.3   |
> |                 | **Ours**       | **4.2±0.2**| **14.1±0.2**| **23.6±0.6**| **8.4±0.2**| **23.1±0.8**| **35.0±1.3**|
> | RegNet+BERT | TESLA-VL       | 3.2±0.8    | 11.1±1.8   | 17.5±1.3   | 5.8±0.1    | 18.6±0.6   | 28.1±1.0   |
> |                 | LoRS           | 3.5±0.1    | 12.6±0.3   | 21.1±0.4   | 6.8±0.3    | 20.8±0.3   | 30.2±0.3   |
> |                 | **Ours**       | **3.9±0.2**| **13.9±0.3**| **24.0±0.6**| **7.9±0.3**| **24.2±0.3**| **36.2±1.1**|
>
>
> ***
> **Q4 (Weaknesses 3):** Some more minor points. 1) Tables 1 and 2 should float to the bottom of the page for readability. 2) Table 4 can also become horizontal instead of vertical, so that it can float to top after Table 3. 3) I would recommend highlighting (bolding / different color) the difference between Equations 5 and 6, since at first glance they are too similar (the difference being the use of image encoder or image projection head). 4) In line 168, “blends” -> “blend”.
>
> **A4:** Thank you for the helpful suggestions. We will (1) adjust the float positions of Tables 1–2, (2) reformat Table 4 horizontally, (3) highlight the difference between Eq. 5 and Eq. 6, and (4) fix the typo “blends” → “blend” in line 168 in the final version.
>
> ***
> **Q5 (Questions 2):**
> Also, I would be grateful if the authors could let me know whether they have tried Representation Blending in the unimodal setting (as it seems that it might be useful there as well).
>
> **A5:** Thank you for the insightful suggestion. We have evaluated the proposed Representation Blending (RB) in the unimodal setting. Specifically, we adopt SRe2L [4] as the baseline and incorporate RB into its distillation process. RB effectively mitigates feature redundancy and encourages the distillation of more class-discriminative representations. The results below show that integrating RB consistently improves the performance of SRe2L across different IPC budgets.
>
> - *Distill CIFAR-100 with ResNet-18*
>
> | IPC | SRe2L [4] | SRe2L + RB |
> | -------- | -------- | -------- |
> | 1     | 7.3     | **7.7**    |
> | 10     | 26.3     | **27.4**     |
> | 50     |  51.6    |  **52.8**    |
>
> *[1] Relaxing Contrastiveness in Multimodal Representation Learning, WACV 2023.*
>
> *[2] Uncovering Meanings of Embeddings via Partial Orthogonality, NeurIPS 2023.*
>
> *[3] Low-Rank Similarity Mining for Multimodal Dataset Distillation, ICML 2024.*
>
> *[4] Squeeze, Recover and Relabel: Dataset Condensation at ImageNet Scale From A New Perspective. NeurIPS 2023.*

---

> > ### Comment · Reviewer_t2RZ · 2025-08-04
> > **Re: Rebuttal**
> >
> > Thank you for the responses to both mine and the other Reviewers' comments. As I mentioned in my earlier review, I think that this is a solid paper. As such, I am keeping my score of "Accept".

---

> > > ### Author Response · Authors · 2025-08-05
> > > **Thank you for your prompt reply.**
> > >
> > > Thank you very much for your prompt reply! Your comments are very valuable for improving our paper. All the suggested results and revisions will be incorporated into the final version. We sincerely appreciate the time and effort you have dedicated.

---

### Official Review · Reviewer_DAj6 · 2025-07-02

**Clarity:** 3
**Significance:** 3
**Originality:** 3
**Rating:** 4
**Confidence:** 3

**Summary:**

This paper addresses the "Modality Collapse" problem which is common in Multimodal Dataset Distillation (MDD), where compressing datasets may lead to low-diversity representations and poor alignment across modalities. This paper proposes RepBlend framework, including two key techniques: 1) Representation Blending (RB): Mixes instance representations (instead of data examples) to enhance intra-modal diversity; 2) Symmetric Projection Trajectory Matching (SM): Balances the optimization process across modalities, by adding an image projection head (with image encoder frozen) and matching optimization trajectories for both modalities symmetrically.

The result shows improvement against previous SOTA methods on Flickr-30K and MS-COCO in image-text retrieval tasks. Moreover, its computational efficiency is also impressive, achieving 6.7x speedup in distillation time by avoiding training the entire image encoder, achieving 6.7x speedup in distillation time (due to frozen image encoder).

Overall, this paper presents a simple, intuitive, yet effective and efficient method for a rapidly evolving domain (though a bit narrow at this moment), which could provide valuable insights for the multimodal research community.

**Questions:**

How many parameters do the models have in the experiments?

**Ethical Concerns:**

["NO or VERY MINOR ethics concerns only"]

**Final Justification:**

My concerns regarding scaling and insufficient evaluations were addressed during the rebuttal process. Considering that 1) the gap between baseline and the distilled data is still non-trivial, and 2) the proposed method could be insightful for the community, I would like to maintain my previous score.

**Limitations:**

yes

**Quality:**

3

**Strengths And Weaknesses:**

Strengths

MDD is a rapidly advancing and promising area of research. This paper identifies the "modality collapse" problem by providing a convincing analysis of its root cause, as illustrated in Figure 1, which depicts how the common methods such as LoRS produce poorly aligned and overly concentrated embeddings. In Figure 2, it further demonstrates how cross-modal supervision reinforces intra-modal similarity and leads to collapse. This problem formulation and explanation are all valuable and insightful.

The proposed solutions RB and SM are elegant and effective. The idea of RB is similar to MixUp but applied in the representation space, which smartly weakens the overly strong cross-modal supervision. SM is to mitigate the asymmetric optimization, by simply freezing the large image encoder and adding a lightweight image projection head.

The efficiency is also largely improved, thanks to the freezing of the image encoder. This leads to 6.7x speedup during the main distillation phase, and reduces peak GPU memory usage by more than half.

This paper is very well written and easy to read.

Weaknesses

It’s unclear whether the method is still effective when scaling the model and data, because all the presented experiments were conducted on small models and datasets. The IR numbers are too small compared to SOTA models (we don’t need to achieve SOTA as well, but better to be more comparable).

Given the proposed method is to reduce the modality gap, demonstrated in image and text embeddings, more zero-shot tasks could be more convincing, such as INET classification, OCR-relevant retrieval (e.g. textcaps) etc.

---

> ### Author Rebuttal · Authors · 2025-07-31
>
> Thank you for your time and constructive comments. We have responded to all your questions as follows and hope these address your concerns.
>
> **Q1 (Weaknesses 1):**
> It’s unclear whether the method is still effective when scaling the model and data, because all the presented experiments were conducted on small models and datasets. The IR numbers are too small compared to SOTA models (we don’t need to achieve SOTA as well, but better to be more comparable).
>
> **A1:** Thank you for raising this important point. To address the concern, we have extended our method to a larger-scale setting using the ***LLaVA-cc3m*** dataset, which serves as the pretraining dataset for LLaVA and consists of 558k image-text pairs. We use approximately 60% of the data (about 334k pairs) for training and reserve a non-overlapping set of 10k pairs for validation. In addition, we evaluate our approach with more powerful encoders, including ***DiNo-v2*** (85,798,656 parameters) for vision and ***BGE-1.5*** (109,482,240 parameters) for text. The following results demonstrate that our method remains effective when scaling both model capacity and training data, while also showing clear superiority over the state-of-the-art competitor. We note that a performance gap against the baseline still exists, which remains a longstanding challenge. Addressing this gap will be a key focus of future work toward achieving lossless or near-lossless distillation.
>
> - *NFNet + Bert on LLaVA-cc3m dataset. The model trained on the full dataset performs: IR@1=9.13, IR@5=25.94, IR@10=36.34, TR@1=9.49, TR@5=26.08, TR@10=37.07.*
>
> | pairs | Methods | IR@1 | IR@5 | IR@10| TR@1 | TR@5 | TR@10
> | -------- | -------- | -------- | -------- | -------- | -------- | -------- | -------- |
> | 199 | LoRS     |   1.56|5.33|8.77|1.01|4.04|6.81
> | | **ours**     | **3.53**     | **12.38**    |**19.33**|**4.39**|**13.45**|**20.18**
> | 499 | LoRS     |1.96|6.72|10.55|1.41|5.11|8.51
> | | **ours**     |**4.45**|**15.09**|**22.31**|**5.14**|**14.89**|**22.97**
> | 799 | LoRS     |1.67   |  5.87    |9.69|1.68|6.11|10.25
> | | **ours** |**5.26**|**16.31**|**24.23**|**5.42**|**16.13**|**24.16**
>
> - *DiNo-v2 + BGE-1.5 on COCO dataset. The model trained on the full dataset performs: IR@1=22.70, IR@5=51.13, IR@10=65.26, TR@1=31.04, TR@5=61.96, TR@10=74.1.*
>
> | pairs| Methods | IR@1 | IR@5 | IR@10| TR@1 | TR@5 | TR@10
> | --------| -------- | -------- | -------- | -------- | -------- | -------- | -------- |
> | 199| LoRS     |1.59|6.12|10.59|1.62|6.15|10.47
> |      | **ours**     |  **12.52**    |**31.96**|**44.39**|**16.06**|**36.84**|**49.34**
> | 499| LoRS     |2.48|8.46|13.42|3.18|10.36|16.37
> |      | **ours**     |  **13.37**    |**33.09**|**45.69**|**16.90**|**39.38**|**52.14**
> | 799 | LoRS    |  2.95     |  9.9    |15.69|4.56|13.66|20.63 |
> | | **ours** |**13.68**|**33.58**|**45.93**|**17.14**|**39.76**|**52.92**
>
> *LLaVA-cc3m: https://github.com/haotian-liu/LLaVA/blob/main/docs/Data.md*
>
> *DiNo-v2: https://huggingface.co/timm/vit_base_patch16_224.dino*
>
> *BGE-1.5: https://huggingface.co/BAAI/bge-base-en-v1.5*
>
> ---
>
> **Q2 (Weaknesses 2):**
> Given the proposed method is to reduce the modality gap, demonstrated in image and text embeddings, more zero-shot tasks could be more convincing, such as INET classification, OCR-relevant retrieval (e.g. textcaps) etc.
>
> **A2:** Thank you for raising this point. To further validate the effectiveness of the reduced modality gap in our distilled dataset, we additionally evaluate zero-shot ***ImageNet*** classification and OCR-relevant retrieval on the ***TextCaps*** dataset [1]. Specifically, we randomly select 10 classes from ImageNet-1K and report the Top-1 and Top-5 zero-shot classification accuracies. For TextCaps, we measure retrieval performance over 3,166 validation samples.
> Here, '*Baseline*' denotes the model trained on the full COCO dataset and directly evaluated on these two tasks in a zero-shot manner. LoRS and Ours correspond to models trained with only 499 distilled COCO image-text pairs. The results, summarized below, show that under the same budget, models trained on our distilled dataset outperform LoRS,  narrowing the performance gap with the full-dataset baseline.
>
> - *ImageNet-10 zero-shot classification*
>
> | Methods     |Acc1 | Acc5
> | -------- | -------- | -------- |
> | Baseline     |   36.4 | 83.2|
> | LoRS    |  21.4 | 74.4|
> | **Ours**    |  **27.6**  | **76.2**|
>
>
> - *TextCaps zero-shot retrieval*
>
> | Methods     |IR@1 | IR@5         | IR@10     |  TR@1  | TR@5         | TR@10
> | -------- | -------- | -------- |-------- | -------- |-------- | -------- |
> | Baseline     |   7.4|18.1  | 26.2   |5.1|14.7|21.3
> | LoRS    |    1.7  |  5.1   |8.4|0.4|1.7|3.1|
> | **Ours**    |   **3.1**   |   **9.4**   |**14.5**|**1.9**|**6.2**|**10.3**
>
> ---
> **Q3 (Questions 1):** How many parameters do the models have in the experiments?
>
> **A3:** Thanks for the question. In our experiments, we have considered four image encoders, including ***ViT, RegNet, ResNet-50***, and ***NFNet***, and two text encoders including ***BERT*** and ***DistilBERT*** (same as baseline LoRS). To evaluate the scalability of our method, we further extended it to more powerful encoders, namely ***DiNo-v2*** for vision and ***BGE-1.5*** for language. The selected models encompass representative architectures with varying capacities, and their parameters are summarized below.
>
> |  | Models | Paras |Official link|
> | -------- | -------- | -------- |-------- |
> |Image encoders |ViT|5,717,416 | https://huggingface.co/WinKawaks/vit-tiny-patch16-224 |
> ||RegNet|9,262,984|https://huggingface.co/timm/nf_regnet_b1.ra2_in1k |
> ||ResNet-50|23,508,032 |https://huggingface.co/microsoft/resnet-50
> |     | NFNet     | 32,769,488     |https://huggingface.co/timm/nfnet_l0.ra2_in1k |
> || DiNo-v2|85,798,656|https://huggingface.co/timm/vit_base_patch16_224.dino|
> | Text encoders     | BERT     |   109,482,240   | https://huggingface.co/google-bert/bert-base-uncased
> ||DistilBERT|66,362,880| https://huggingface.co/distilbert/distilbert-base-uncased
> ||BGE-1.5|109,482,240|https://huggingface.co/BAAI/bge-base-en-v1.5|
>
> The superiority demonstrated across these diverse models not only validates the effectiveness of the proposed method, but also highlights its scalability and generalization capability.
>
> *[1] TextCaps: a Dataset for Image Captioning with Reading Comprehension, ECCV 2020.*

---

> > ### Comment · Reviewer_DAj6 · 2025-08-05
> >
> > Thank you for the detailed explanation, which has addressed most of my concerns.
> >
> > > We note that a performance gap against the baseline still exists, which remains a longstanding challenge.
> >
> > Where can I find the baseline numbers?

---

> > > ### Author Response · Authors · 2025-08-06
> > > **Thank you very much for your prompt reply.**
> > >
> > > Thank you very much for your prompt reply. We are glad to hear that most of your concerns have been addressed. Your comments have been very helpful in improving our paper.
> > >
> > > Regarding the baseline numbers, they represent the performance of models trained on the original full datasets. As noted in the table captions: for NFNet + BERT on the LLaVA-cc3m dataset, the model achieves IR@1 = 9.13, IR@5 = 25.94, IR@10 = 36.34, TR@1 = 9.49, TR@5 = 26.08, and TR@10 = 37.07; and for DINO-v2 + BGE-1.5 on the COCO dataset, the model achieves IR@1 = 22.70, IR@5 = 51.13, IR@10 = 65.26, TR@1 = 31.04, TR@5 = 61.96, and TR@10 = 74.10. Approaching such baseline performance with a distilled training dataset remains a longstanding and challenging problem.

---

> > > > ### Comment · Reviewer_DAj6 · 2025-08-06
> > > >
> > > > I appreciate the authors' clarifications, which have resolved my previous questions. I am happy to maintain my score.

---

> > > > > ### Author Response · Authors · 2025-08-07
> > > > > **Thank you for your time and thoughtful review.**
> > > > >
> > > > > Thank you once again for your time and thoughtful review. We sincerely appreciate your constructive feedback, which has helped us clarify and improve the paper. We will ensure that the corresponding revisions are incorporated into the final version.

---

### Official Review · Reviewer_bSSh · 2025-07-02

**Clarity:** 2
**Significance:** 3
**Originality:** 3
**Rating:** 4
**Confidence:** 3

**Summary:**

### **Strengths**

1. **Problem Significance**:
   - First to formally define and analyze *modality collapse* in MDD, linking it to over-compression and contrastive supervision (Sec. 3.1). Theoretical analysis (Prop. in Sec. 3.1, Appendix B) rigorously justifies the cause.
2. **Innovative Solutions**:
   - **Representation Blending**: Effectively diversifies intra-modal features via in-distribution MixUp-like blending (Eq. 5, Fig. 3), outperforming naive noise injection.
   - **Symmetric Projection Matching**: Resolves asymmetric supervision (Fig. 4) by aligning image/text projection trajectories (Eq. 6), improving cross-modal alignment.
3. **Strong Empirical Validation**:
   - RepBlend consistently outperforms SOTA (LoRS, MTT-VL) across datasets (Tables 1–2), metrics (e.g., +30% TR@10 on Flickr-30K at 500 pairs), and modalities (audio-text in Table 7).
   - Ablations (Fig. 5) confirm each component’s contribution; cross-architecture tests (Table 3, Fig. 6–7) show robustness.
4. **Efficiency Gains**:
   - Achieves 6.7× speedup and 2.14× memory reduction via lightweight projection-head matching (Table 4).

------

### **Weaknesses**

1. Altough the authors propose an informable algorithmatic issue, the modality collapas, it would be more clear if the authors could provide a detailed example to facilitate the reader's understanding.
2. The relationship between Section 3.2 and 3.3 is not clearly stated. Are they two different ways of method implementation? or do they need be utilized simultaneously?
2. **Visualization Insufficiency**:
   - Fig. 9–10 show distilled images/texts but lack quantitative or qualitative analysis (e.g., diversity metrics, attention maps). Claims about "enhanced diversity" need stronger visual proof.
3. Since LoRS is an important baseline throughout the experiment section, there should be explicit definition of LoRS in the formal part of paper.
4. The structure of Section 3.3 can also be improved. It may be better to use a prelimiary section to introduce LoRS, while the method section should focus on the different between LoRS and RepBlend.

------

**Questions:**

1. Although the authors propose an informative algorithmic issue, the modality collapses, it would be clearer if the authors could provide a detailed example to facilitate the reader's understanding.
2. The relationship between Section 3.2 and 3.3 is not clearly stated. Are they two different ways of method implementation? or do they need to be utilized simultaneously?
2. **Visualization Insufficiency**:
   - Fig. 9–10 show distilled images/texts but lack quantitative or qualitative analysis (e.g., diversity metrics, attention maps). Claims about "enhanced diversity" need stronger visual proof.
3. Since LoRS is an important baseline throughout the experiment section, there should be an explicit definition of LoRS in the formal part of paper.
4. The structure of Section 3.3 can also be improved. It may be better to use a preliminary section to introduce LoRS, while the method section should focus on the difference between LoRS and RepBlend.

**Ethical Concerns:**

["NO or VERY MINOR ethics concerns only"]

**Final Justification:**

Most of the concerns are addressed. We believe this is a reasonable paper and will keep our positive score.

**Limitations:**

yes

**Quality:**

3

**Strengths And Weaknesses:**

### **Strengths**

1. **Problem Significance**:
   - First to formally define and analyze *modality collapse* in MDD, linking it to over-compression and contrastive supervision (Sec. 3.1). Theoretical analysis (Prop. in Sec. 3.1, Appendix B) rigorously justifies the cause.
2. **Innovative Solutions**:
   - **Representation Blending**: Effectively diversifies intra-modal features via in-distribution MixUp-like blending (Eq. 5, Fig. 3), outperforming naive noise injection.
   - **Symmetric Projection Matching**: Resolves asymmetric supervision (Fig. 4) by aligning image/text projection trajectories (Eq. 6), improving cross-modal alignment.
3. **Strong Empirical Validation**:
   - RepBlend consistently outperforms SOTA (LoRS, MTT-VL) across datasets (Tables 1–2), metrics (e.g., +30% TR@10 on Flickr-30K at 500 pairs), and modalities (audio-text in Table 7).
   - Ablations (Fig. 5) confirm each component’s contribution; cross-architecture tests (Table 3, Fig. 6–7) show robustness.
4. **Efficiency Gains**:
   - Achieves 6.7× speedup and 2.14× memory reduction via lightweight projection-head matching (Table 4).

------

### **Weaknesses**

1. Although the authors propose an informative algorithmic issue, the modality collapses; it would be clearer if the authors could provide a detailed example to facilitate the reader's understanding.
2. The relationship between Section 3.2 and 3.3 is not clearly stated. Are they two different ways of method implementation? or do they need to be utilized simultaneously?
2. **Visualization Insufficiency**:
   - Fig. 9–10 show distilled images/texts but lacks quantitative or qualitative analysis (e.g., diversity metrics, attention maps). Claims about "enhanced diversity" need stronger visual proof.
3. Since LoRS is an important baseline throughout the experiment section, there should be an explicit definition of LoRS in the formal part of paper.
4. The structure of Section 3.3 can also be improved. It may be better to use a preliminary section to introduce LoRS, while the method section should focus on the difference between LoRS and RepBlend.

---

> ### Author Rebuttal · Authors · 2025-07-31
>
> Thank you for your constructive comments! We give point-to-point replies to your questions in the following.
>
> **Q1 (Weaknesses 1 & Questions 1):** Although the authors propose an informable algorithmatic issue, the modality collapse, it would be more clear if the authors could provide a detailed example to facilitate the reader's understanding.
>
> **A1:** Thank you for the question.
>  - Definition of ***Modality Collapse***
> In our paper, *modality collapse* refers to a phenomenon in multimodal dataset distillation (MDD) where the distilled embeddings within the same modality become overly concentrated (i.e., exhibit low intra-modal variance), while the alignment between cross-modal embeddings (e.g., image-text pairs) is insufficient. This is visually illustrated in ***Figure 1***, where intra-modal representations cluster tightly, yet fail to align across modalities.
> - Detrimental effect of ***Modality Collapse***
> Such collapse results in degraded representational diversity and weakened multimodal correspondence (as shown in ***Figure 2***), ultimately leading to significant performance degradation when models are trained on the distilled data, compared to those trained on the original data, as quantitatively demonstrated in Tables I and II.
> ---
> **Q2 (Weaknesses 2 & Questions 2):** The relationship between Section 3.2 and 3.3 is not clearly stated. Are they two different ways of method implementation? or do they need be utilized simultaneously?
>
> **A2:** Thank you for the question. Representation Blending (Section 3.1) provides a theoretically sound solution to mitigate modality collapse by encouraging intra-modal diversity. While this strategy can effectively reduce intra-modal concentration and slightly narrow the cross-modal gap (from 0.327 in LoRS to 0.318), its impact remains limited due to the inherently asymmetric nature of the distillation process.
>
> To address this asymmetry and further enhance cross-modal alignment, we introduce a Symmetric Projection Matching strategy (Section 3.2), which aligns the projection-head trajectories of both modalities during distillation. This method significantly reduces the modality gap (from 0.327 in LoRS to 0.230), demonstrating its effectiveness in improving cross-modal alignment.
>
> Combining these two components achieves a synergistic effect: Representation Blending enhances intra-modal diversity, while Symmetric Projection Matching ensures cross-modal alignment, thereby significantly improving the quality of the distilled dataset.
>
> ---
>
> **Q3 (Weaknesses 3 & Questions 3):** Fig. 9–10 show distilled images/texts but lack quantitative or qualitative analysis (e.g., diversity metrics, attention maps). Claims about "enhanced diversity" need stronger visual proof.
>
> **A3:** Thank you for the constructive suggestion.
> - Quantitative results:
> The enhanced diversity of our distilled dataset is reflected in two aspects: **(1) reduced intra-modal similarity**, and **(2) a narrower inter-modal gap**. We quantify them using the **Concentration Ratio (CR)**, **intra-modal similarity (Sim)**, and **modality gap (Gap)** as defined in the main paper. The following tables summarize these quantitative results, showing that our method substantially improves diversity compared to LoRS.
>
> | Metric | LoRS | Ours |
> |--------|------|------|
> | **CR (Sim)** – *Image* | 0.95 (0.522) | **0.57 (0.512)** |
> | **CR (Sim)** – *Text*  | 0.888 (0.519) | **0.45 (0.5104)** |
>
> | Metric | LoRS | Ours |
> |--------|------|------|
> | **Gap** - *(Image ↔ Text)* | 0.327 | **0.230** |
>
> - Qualitative Visualizations:
> In addition to the quantitative results, **the qualitative visualizations in Figure 1** clearly demonstrate that the distilled embeddings are more dispersed within modalities and better aligned across modalities. Furthermore, **the right subfigure in Figure 2** depicts a cross-modal similarity heatmap (functionally equivalent to an attention map), where each image exhibits distinct attention patterns toward the text, similarly, each text also attends differently to the images. This highlights the increased instance-wise diversity in cross-modal interactions.
>
> As figures are not permitted during the rebuttal, we will revise the final version to better highlight and elaborate on these visualizations and their implications, and provide further qualitative illustrations to substantiate our claims.
>
> ---
>
> **Q4 (Weaknesses 4 & 5 & Questions 4 & 5):** Since LoRS is an important baseline throughout the experiment section, there should be explicit definition of LoRS in the formal part of paper. The structure of Section 3.3 can also be improved. It may be better to use a prelimiary section to introduce LoRS, while the method section should focus on the different between LoRS and RepBlend.
>
> **A4:** Thank you for the valuable suggestions. In the current version, Section 2 (Preliminaries and Related Work) introduces the Multimodal Dataset Distillation (MDD) problem, contextualized by two representative baselines: LoRS and MTT-VL. The details of LoRS are also described at the beginning of Section 3.1.
> To improve clarity and coherence, we have restructured these sections by explicitly formalizing LoRS in the preliminaries (see below), while leaving the method section to focus on the distinctions between LoRS and RepBlend.
>
> -  Preliminaries and Related Works.
>
> Given a large-scale image-text dataset $\mathcal{D} = \{(\mathbf{x}\_i, \boldsymbol{\tau}\_i), \mathbf{y}\_i\}_{i=1}^{|\mathcal{D}|}$,
> where $\mathbf{x}\_i \in \mathbb{R}^{d{\_\text{img}}}$ and $\boldsymbol{\tau}\_i \in \mathbb{R}^{d{\_\text{text}}}$ denote the $i$-th image and its paired caption representation, and each pair is independently sampled from a natural data distribution $\mathcal{P}$.
>
> Each $\mathbf{y}_i \in \lbrace 0,1 \rbrace^{|\mathcal{D}|}$ is a one-hot vector indicating the correspondence between $\mathbf{x}\_i$ and the caption set ${\boldsymbol{\tau}\_j}\_{j=1}^{|\mathcal{D}|}$, with the $i$-th entry activated.
> Similar to Dataset Distillation (DD), Multimodal Dataset Distillation (MDD) also aims to minimize the loss on the original dataset using the model trained on its distilled synthetic counterpart $\mathcal{S} = \{(\tilde{\mathbf{x}}\_i, \tilde{\boldsymbol{\tau}}\_i), \tilde{\mathbf{y}}\_i \}\_{i=1}^{|\mathcal{S}|}$:
> $$\mathcal{S}^* = \arg\min\_{\mathcal{S}} \mathbb{E}\_{(\mathbf{x},\boldsymbol{\tau}) \sim \mathcal{P}} \left[ \mathcal{L}(f\_{\boldsymbol{\theta}\_{\mathcal{S}}}(\mathbf{x}, \boldsymbol{\tau}), \mathbf{y}) \right]\quad \text{s.t.} \quad \boldsymbol{\theta}\_{\mathcal{S}} = \arg\min\_{\boldsymbol{\theta}} \mathbb{E}\_{(\tilde{\mathbf{x}},\tilde{\boldsymbol{\tau}}) \sim \mathcal{S}} \left[ \mathcal{L}(f\_{\boldsymbol{\theta}}(\tilde{\mathbf{x}}, \tilde{\boldsymbol{\tau}}), \tilde{\mathbf{y}}) \right]  [1]
> $$
> where $|\mathcal{S}| \ll |\mathcal{D}|$, and $\mathcal{L}$ denotes the contrastive learning loss. The model $f\_{\boldsymbol{\theta}}(\cdot)$ represents a CLIP-style network parameterized by $\boldsymbol{\theta}$. Each distilled sample consists of a synthetic image-text pair $(\tilde{\mathbf{x}}\_i, \tilde{\boldsymbol{\tau}}\_i)$, where $\tilde{\mathbf{x}}\_i \in \mathbb{R}^{d\_\text{img}}$ and $\tilde{\boldsymbol{\tau}}\_i \in \mathbb{R}^{d\_\text{text}}$, accompanied by a learned soft label $\tilde{\mathbf{y}}\_i$.
>
> LoRS is a representative MDD method built upon Equation [1], where the loss function $\mathcal{L}$ is defined as:
> $$\mathcal{L}\_{\mathrm{wBCE}}^{\mathcal{B}} = \sum\_{i,j}^{|\mathcal{B}|} w\_{ij} \cdot \ell\left(\tilde{\mathbf{y}}\_{ij}, \sigma\left(\hat{\mathbf{y}}\_{ij} / \gamma\right)\right), \quad w\_{ij} = \frac{\mathbb{I}[\tilde{\mathbf{y}}\_{ij} > \beta]}{|{(i,j): \tilde{\mathbf{y}}\_{ij} > \beta}|} + \frac{\mathbb{I}[\tilde{\mathbf{y}}\_{ij} \leq \beta]}{|{(i,j): \tilde{\mathbf{y}}\_{ij} \leq \beta}|}.
> $$
> Here, $\mathcal{B} \subset \mathcal{S}$ denotes a sampled batch. $\hat{\mathbf{y}}\_{ij}$ represents the cosine similarity between the normalized image and text embeddings, where: $\tilde{\mathbf{x}}\_i^\prime = \operatorname{Normalize}(f^{\text{imgE}}(\tilde{\mathbf{x}}\_i))$ and $\tilde{\boldsymbol{\tau}}\_j^\prime = \operatorname{Normalize}(f^{\text{textP}}(\tilde{\boldsymbol{\tau}}\_j))$ with $f^{\text{imgE}}(\cdot)$ and $f^{\text{textP}}(\cdot)$ denoting the image encoder and text projection head, respectively. Note that In LoRS [1], no image projection head is used. The threshold $\beta$ is used to determine positive and negative pairs, $\sigma(\cdot)$ denotes the sigmoid function, and $\gamma$ is the temperature. $\ell(\cdot, \cdot)$ refers to the binary cross-entropy loss. While this supervision primarily aims to mine cross-modal relationships, it inadvertently reinforces intra-modal similarities, ultimately leading to Modality Collapse.
>
> *[1] Low-Rank Similarity Mining for Multimodal Dataset Distillation, ICML 2024.*

---

> > ### Comment · Reviewer_bSSh · 2025-08-06
> >
> > Thanks for your clarifications. We notice our Q3, 4, 5 have been resolved. For Q1, what we originally mean is to provide a detailed example (a detailed figure and a text, and their embedding results which have the modality collapse issue). For Q2, we suggest to rename Section 3.1 (to indicate it is the solution of modality collapse, instead of discussing modality collapse itself), and make the logical relationship between 3.1, 3.2 and 3.3 more explicitly. Overall, we believe this is a reasonable paper and will keep our positive score.

---

> > > ### Author Response · Authors · 2025-08-06
> > > **Thank you for your prompt reply.**
> > >
> > > Thank you for your prompt reply. We really appreciate your constructive feedback, which has helped us improve the paper.
> > >
> > > For Q1, we will include a detailed figure–text pair along with the corresponding embedding results in Figure 1, to provide a clearer and more intuitive illustration. For Q2, Section 3.1 was originally intended to introduce the concept of “modality collapse” and analyze its underlying causes and detrimental effects; we will move this part to the Preliminaries section so that Section 3 can focus exclusively on the proposed solutions. Sections 3.2 and 3.3 present our solutions to address modality collapse, and we will refine their presentation to make the logical flow more explicit.
> > >
> > > Thanks again for the time and effort you dedicated to the review.

---

### Note · Authors · 2025-08-13

Dear AC,

We would like to express our sincere gratitude to all reviewers for their thoughtful evaluations and encouraging feedback on our submission. We are pleased that the reviewers recognized its good presentation, novelty, solid theoretical foundation, and convincing results, with the work receiving an **initial all-positive rating of 4-4-5-4**.

We also appreciate the reviewers’ constructive engagement during the rebuttal phase, which prompted us to clarify key points, refine the presentation, and conduct additional supporting experiments. **All concerns were addressed to the reviewers’ satisfaction**, and the resulting revisions and supplementary experiments will be incorporated into the final version.

Notably, **all reviewers indicated that they were happy to maintain their positive ratings** after the discussion phase, reflecting a shared view that the work meets the high standards expected for acceptance.

Once again, we sincerely thank the reviewers and AC for the time, effort, and thoughtful consideration they devoted to reviewing this work.


Authors of submission 7553

---

### Decision · Program_Chairs · 2025-09-17

**Decision:**

Accept (poster)

**Comment:**

The paper addresses Modality Collapse in Multimodal Dataset Distillation (MDD). Modality Collapse is a phenomenon in which  dataset compression leads to low diversity and poor alignment across modalities. The authors propose RepBlend, which consists of two components: Representation Blending (RB) to enhance intra-modal diversity by mixing instance representations, and Symmetric Projection Trajectory Matching (SM) to balance optimization across modalities by using a frozen image encoder. RepBlend shows superior performance in image-text retrieval tasks, improving over state-of-the-art methods while achieving a 6.7x speedup in distillation time by not requiring the training of the entire image encoder.

Reviewers generally agree that this paper makes significant contributions by being the first to formally define and analyze Modality Collapse in Multimodal Dataset Distillation (MDD), and by linking it to over-compression and contrastive supervision. They find that the paper introduces novel solutions like Representation Blending, which diversifies intra-modal features effectively, and Symmetric Projection Trajectory Matching, which resolves asymmetric supervision to improve cross-modal alignment. Reviewers recognize the strong empirical validation, as the proposed method consistently outperforms state-of-the-art techniques across multiple datasets and modalities. Additionally, reviewers appreciate the substantial efficiency gains achieved (6.7x speedup and significant memory reduction) thanks to lightweight projection-head matching. Finally, reviewers unanimously praise the writing and technical exposition as being clear and well-supported.

Despite the generally positive initial opinions from the reviewers, paper has several areas pointed out for improvement:

+ The lack of explicit definition for the baseline method LoRS and insufficient visualization, such as diversity metrics or attention maps, weakens the claims about enhanced diversity. In rebuttal, the authors provided a quantitative comparison between LoRS and the proposed approach based on the Concentration Ration, demonstrating superior diversity after distillation.
+ The experiments are conducted only on small models and datasets, which raised concerns about scalability, and that there is a lack of zero-shot tasks to showcase the effectiveness. In rebuttal, the authors provided a number of new experimental results on different datasets, with larger models, and on zero-shot tasks. These results further demonstrate the effectiveness of the proposed approach and the importance of modality collapse.

The reviewers were satisfied with the author clarifications and rebuttals, and in summary are in agreement that both main ideas presented are solid and novel, with experimental evaluation convincingly demonstrating the superior performance of the proposed approach. The recommendation is thus to Accept.